# Rel-MOSS: Towards Imbalanced Relational Deep Learning on Relational Databases

**Jun Yin** [1]  **Peng Huo** [2]  **Bangguo Zhu** [3]  **Hao Yan** [3]  **Senzhang Wang** [3]  **Shirui Pan** [4]  **Chengqi Zhang** [1]

## Abstract

In recent advances, to enable a fully data-driven learning paradigm on relational databases (RDB), relational deep learning (RDL) is proposed to structure the RDB as a heterogeneous entity graph and adopt the graph neural network (GNN) as the predictive model. However, existing RDL methods neglect the imbalance problem of relational data in RDBs and risk under-representing the minority entities, leading to an unusable model in practice. In this work, we investigate, for the first time, class imbalance problem in RDB entity classification and design the relation-centric minority synthetic over-sampling GNN (**Rel-MOSS**), in order to fill a critical void in the current literature. Specifically, to mitigate the issue of minority-related information being submerged by majority counterparts, we design the relation-wise gating controller to modulate neighborhood messages from each individual relation type. Based on the relational-gated representations, we further propose the relation-guided minority synthesizer for over-sampling, which integrates the entity relational signatures to maintain relational consistency. Extensive experiments on 12 entity classification datasets provide compelling evidence for the superiority of Rel-MOSS, yielding an average improvement of up to 2.46% and 4.00% in terms of *Balanced Accuracy* and *G-Mean*, compared with SOTA RDL methods and classic methods for handling class imbalance problem.

[1]Department of Data Science and Artificial Intelligence, Hong Kong Polytechnic University, Hong Kong SAR, China [2]National Super Computing Center, Tianjin, China [3]School of Computer Science and Engineering, Central South University, Hunan, China [4]School of Information and Communication Technology, Griffith University, Queensland, Australia. Correspondence to: Shirui Pan <s.pan@griffith.edu.au>, Chengqi Zhang <Chengqi.zhang@polyu.edu.hk>.

*Proceedings of the 43rd International Conference on Machine Learning*, Seoul, South Korea. PMLR 306, 2026. Copyright 2026 by the author(s).

## 1. Introduction

Relational databases (RDB), which consist of multiple inter-connected tables via primary-foreign key relations, are the most widely used database management systems, spanning e-commerce, social media, and healthcare (Garcia-Molina, 2008; Fey et al., 2024; Robinson et al., 2024; Wang et al., 2024). Currently, to fully exploit the predictive signal encoded in the relations between database entities, *relational deep learning* (RDL) is proposed to re-cast the relational data as a heterogeneous entity graph and enable end-to-end optimization (Fey et al., 2024), regardless of laborious feature engineering (Ke et al., 2017).

Despite the remarkable performance of existing RDL methods (Robinson et al., 2024; Hu et al., 2020; Chen et al., 2025) on RDB predictive tasks, they mostly overlook the imbalance problem, which is ubiquitous in real-world data sources (Zhao et al., 2021; Wang et al., 2020; Ma et al., 2025). Especially for the RDB entity classification task (such as fake account detection (Mohammadrezaei et al., 2018) and customer churn prediction (Ni et al., 2019)), which plays a fundamental role in real-world applications, the consequences of overlooking the class imbalance problem are particularly severe. With the fake account detection task as an example, the overwhelming majority of user accounts are benign, while fraudulent accounts occupy only a small portion. If a classification model is trained without addressing the class imbalance problem, it may completely miss the minority instances that are actually critical and label every instance as *benign*, inflicting immeasurable losses.

Although countermeasures for the class imbalance problem have been extensively studied in graph data-mining (Zhao et al., 2021; Wu et al., 2022), previous methods primarily focus on homogeneous graphs with relatively simple connectivity patterns (Ma et al., 2025). When applied to heterogeneous entity graphs derived from RDBs, the following challenges caused by complex relational structures must be addressed with particular care. ***First, different types of relations vary greatly in the extents they contribute to exploiting minority-related information.*** As shown in Figure 1a), compared to the majority, minority entities display distinct relational connectivity patterns, while suffering from substantially weaker connection strengths (Wang et al.,

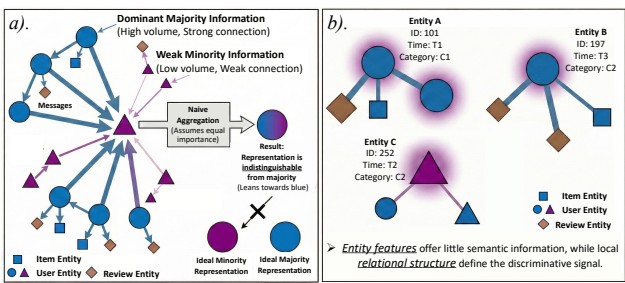

*Figure 1.* Illustration of *a).* Aggregation with equal importance leads to indistinguishable representations and *b)* Decisive information resides in local relational structures.

2022). When messages from various relations are assumed to be equally important, information from the majority class dominates the message-passing process, resulting in representations of the minority and majority classes that become nearly indistinguishable. ***Second, the decisive information relevant to RDB entity classification implicitly resides in the local relational structures formed by multiple interconnected entities.*** As illustrated by Figure 1*b)*, in RDBs, the features of an entity (typically numeric, categorical, and timestamp) mainly indicate the entity identification, which is less semantically informative compared to node features in common graphs (Yan et al., 2023a; 2025). Therefore, when dealing with imbalanced entity classification, preserving and leveraging the relational structures is essential for distinguishing the decision boundary.

In this work, we propose the relation-centric minority synthetic over-sampling GNN (**Rel-MOSS**), which is intensely grounded in the relational structures within heterogeneous entity graphs. In particular, Rel-MOSS is composed of a relation-wise gating controller (Rel-Gate) and a relation-guided minority synthesizer (Rel-Syn), which are respectively designed to tackle the two aforementioned challenges. Specifically, for the input RDB entity, Rel-Gate first estimates, for each relation type, the likelihood of neighbor messages leaning towards the minority class. Afterwards, Rel-Gate enhances the minority-relevant information while restraining the opposing information according to the estimated likelihood, thereby making the representations of minority and majority entities more easily distinguishable. Based on the gated entity representation, Rel-Syn extends the minority over-sampling technique to heterogeneous entity graphs. By integrating diverse structural statistics of the entity neighborhood as relational signatures, Rel-Syn is able to maintain relational consistency and thereby synthesize faithful minority samples. Our main contributions are summarized as follows.

• We investigate, for the first time, the class imbalance problem of entity classification on RDBs. Accordingly, we propose the relation-centric minority synthetic over-sampling GNN (Rel-MOSS), strongly anchored in the relational struc-

tures of heterogeneous entity graphs.

• We design a relation-wise gating controller (Rel-Gate) that adaptively modulates the message passing procedure for each relation, thereby rendering the distinction between the representations of minority and majority entities more pronounced. We also propose Rel-Syn, which augments the minority synthesizer with the guidance of relational signatures, allowing it to preserve relational consistency.

• Experiments on 12 entity classification benchmarks provide compelling evidence for the superiority of Rel-MOSS, which achieves an average improvement of up to 2.46% and 4.00% in terms of *Balanced Accuracy* and *G-Mean*.

## 2. Related Work

**Relational Deep Learning.** In order to directly learn from relational databases (RDB) (Kanter & Veeramacha-neni, 2015; Cvetkov-Iliev et al., 2023) consisting of multiple connected tables, Fey et al. (2024) introduce relational deep learning (RDL), an end-to-end deep learning paradigm for RDBs. RDL evolves from manual feature engineering and handcrafted systems to fully data-driven representation learning systems, by representing a relational database as a heterogeneous relational entity graph (Fey et al., 2024). Subsequently, GNNs can be applied to build end-to-end data-driven predictive models. Hitherto, a growing corpus of studies has been developed within the framework of RDL (Wang et al., 2025; Ranjan et al., 2025).

Most of them are committed to advanced architectures, including heterogeneous GNNs, expressive graph transformers (Hu et al., 2020), meta-path inspired message passing module (Chen et al., 2025), local-global information exchanging module (Dwivedi et al., 2025). However, they neglect the class imbalance, one of the most substantial problems in the field of machine learning, especially when handling entity classification on RDBs.

**Class Imbalance Problem.** Class imbalance is common in real-world applications and has long been a classical research direction. Classes with a larger number of instances are usually called *majority* classes, while those with fewer instances are usually called *minority* classes. The countermeasures can be roughly classified into data-level and learning-algorithm-level methods. The most popular data-level method, SMOTE (Chawla et al., 2002), addresses this problem by performing interpolation between samples in minority classes and their nearest neighbors. Many extensions on top of SMOTE are proposed to make the interpolation process more effective (Han et al., 2005; Bunkhumpornpat et al., 2009). At the learning algorithm level, cost sensitive learning (Lin et al., 2017) constructs a cost matrix to assign different mis-classification penalties for different classes. Post-hoc adjustment (Yun et al., 2022) method modifies the

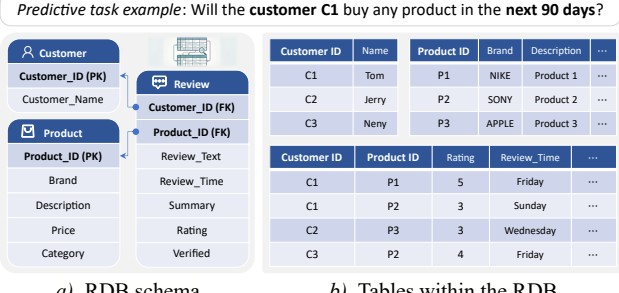

*Predictive task example*: Will the **customer C1** buy any product in the **next 90 days**?

*a).* RDB schema        *b).* Tables within the RDB

*Figure 2.* Illustrative example of *a).* Relational database schema, and *b).* Tables within the relational database.

inference process after the classifier is trained, by introducing a prior probability for each class.

In the field of *graph data-mining*, class-imbalanced learning is an emerging research area addressing class imbalance in graph data (Zhao et al., 2021; Wu et al., 2022; Li et al., 2023; 2024), where traditional methods for non-graph data might be unsuitable or ineffective. Several graph-specific adaptations have been proposed. GraphSMOTE (Zhao et al., 2021) is the first to adapt SMOTE for graphs by performing interpolation in the node embedding space and using a pretrained edge predictor to determine the connectivity between synthetic and real nodes. GraphMixup (Wu et al., 2022) performs mixup in the semantic space to prevent the generation of out-of-domain samples and incorporates self-supervised objectives to preserve node connections. GraphSHA (Li et al., 2023) expands minority class boundaries through Mixup between hard minority samples and neighboring nodes using selective edges sampled from the subgraph of minority nodes. GNN-CL (Li et al., 2024) further advances the field through curriculum learning to gradually expose the model to increasingly complex synthetic samples.

Nevertheless, existing class-imbalanced learning methods are primarily limited to homogeneous graphs, whose connectivity patterns are relatively simplistic (Ma et al., 2025) and whose node features are semantically informative. The complex relational structures of heterogeneous RDB entity graphs incur unique challenges for previous methods and thus motivate the design of a novel methodology tailored for imbalanced entity classification on RDBs.

## 3. Preliminary

As illustrated in Figure 2, a *relational database* (RDB) $\mathcal{R} = (\mathcal{T}, \mathcal{S})$ consists of a set of tables $\mathcal{T} = \{T_1, T_2, \cdots, T_{|\mathcal{T}|}\}$ and a set of links between them $\mathcal{S} \subseteq \mathcal{T} \times \mathcal{T}$. Generally, each *table* $T$ can be structured as a matrix with $N$ rows and $M$ columns, where each row $e \in T$ is considered an *entity* with $M$ features. Specifically, each table $T$ has a special column set $\mathcal{P}$ called the *primary key* (PK) that can identify each row within it and distinguish it from other tables. Moreover, each

table can possess several columns $\mathcal{F} = \{f_1, f_2, \cdots, f_{|\mathcal{F}|}\}$ called the *foreign keys* (FKs), which are the primary keys of other tables. The collection of foreign-primary key linkages therefore defines the relational structure of a RDB.

The heterogeneous entity graph derived from a RDB is formally represented as,

$$G = \left( \cup_t \mathcal{V}^t, \cup_r \mathcal{E}^r \right), \tag{1}$$

where the $t$-th node set $\mathcal{V}^t$ is constructed from the $t$-th table $T_t$, and each node in $\mathcal{V}^t$ corresponds to an entity $e \in T_t$. The $r$-th edge set $\mathcal{E}^r$ is constructed according to the $r$-th relationship in the collection of foreign-primary linkages, and each edge connects the source entity from the table with the FK to the referenced entity in the table with the PK. *In this work, we focus on **entity binary classification**, which is prevalent in RDB-related applications.*

## 4. Methodology

In this section, we introduce the relation-centric minority synthetic over-sampling GNN (**Rel-MOSS**) in detail. Figure 3 illustrates the overall architecture. In addition to the modality-specific encoder that converts original features into continuous embeddings, Rel-MOSS is composed of two key modules centered on the relational structures, i.e., the relation-wise gating controller (Rel-Gate) and the relation-guided minority synthesizer (Rel-Syn).

Specifically, for the neighborhood message extracted by the GNN backbone from each relation, Rel-Gate first estimates the likelihoods towards the minority class, thereby modulating each type of message accordingly for discriminative entity representations. Afterwards, Rel-Syn generates faithful samples based on the gated representations. During the generation, the relational signatures of minority samples are integrated to facilitate the utilization of structural information. Along with the classification objective, Rel-MOSS further introduces a reconstruction objective for the relational signatures, in order to preserve the relational consistency.

### 4.1. Modality-specific Feature Encoder

In real-world applications, the entity features in RDBs may cover various modalities (Wang et al., 2024; Robinson et al., 2024), including numeric feature, categorical feature, timestamp, textual content, and even visual images. As shown by the left part of Figure 3, we adopt a dedicated encoder $E_k$ for each modality $k$ to convert the original features into vectorized representations (Hu et al., 2024), following the standard paradigm (Fey et al., 2024). Formally, given an entity containing $M$ features $e = [e_1, e_2, \cdots, e_M]$, the modality specific feature encoding process can be represented as,

$$X_e = \text{Concat}\left(\{E_{k_m}(e_m)\}\right), \tag{2}$$

where $\mathrm{Concat}(\cdot)$ is the *concatenation* operator and $k_m$ denotes the modality corresponding to the $m$-th entity feature. Since entities from different tables can possess distinct feature sets, the RDL pipeline attaches a separate projection MLP to each entity in order to standardize the final dimensionality as $X_e \leftarrow \mathrm{MLP}(X_e) \in \mathbb{R}^d$.

## 4.2. Rel-Gate: Relation-wise Gating Controller

Based on vectorized entity representations, standard heterogeneous GNNs (Hamilton et al., 2017; Wang et al., 2019; Hu et al., 2020) first extract neighborhood messages for each relation type and then aggregate them indiscriminately, which can be briefly formulated as follows.

$$X_e \leftarrow \sigma\left( W_e X_e + \sum_r \frac{1}{|\mathcal{N}_r(e)|} \cdot \sum_{v \in \mathcal{N}_r(e)} W_r X_v \right), \quad (3)$$

where $r$ represents the relation type and $\mathcal{N}_r(e)$ denotes the neighbors of $e$ connected via relation $r$. Since the quantity of majority samples overwhelmingly outnumbers that of minorities, the predictive patterns towards minorities are submerged in the majority information with substantial volumes. The standard message passing process further intensifies the information imbalance and eventually leads to the collapse of minority information.

> **Proposition 4.1** (Minority Information Collapse).
> *WLOG, consider a relational entity graph where entity classification is performed on a single entity type with binary labels $y \in \{0,1\}$, and assume that the minority entities are severely less than the majority ones. According to formula 3, for any minority entity $e$, the magnitude of its minority-discriminative signal decays monotonically across layers, with a contraction factor proportional to the expected minority proportion in its relational neighborhoods.*

Let $X_e^{(l)} \in \mathbb{R}^d$ denote the representation of entity $e$ at layer $l$. We characterize minority information through a conditional difference. For the given entity $e$, define the minority discriminative signal at layer $l$ as follows,

$$\Delta_e^{(l)} = \mathbb{E}\left[ X_v^{(l)} \mid v \in \mathcal{N}(e), y_v \in \mathrm{Minor} \right] \\ - \mathbb{E}\left[ X_v^{(l)} \mid v \in \mathcal{N}(e), y_v \in \mathrm{Major} \right]. \quad (4)$$

Considering the class imbalance and relational connectivity, the proportion of minority neighbors under relation $r$ satisfies the inequality below,

$$\pi_{e,r} = \mathbb{P}\left[ y_v \in \mathrm{Minor} \mid v \in \mathcal{N}_r(e) \right] \ll 0.5. \quad (5)$$

Ignoring the nonlinearity $\sigma(\cdot)$ under a standard local Lipschitz assumption (Virmaux & Scaman, 2018), the expected

update of $X_e^{(l+1)}$ can be decomposed as follows,

$$\mathbb{E}[X_e^{(l+1)}] = \\ \sum_r W_r^{(l)} \left( \pi_{e,r} \cdot \mu_{e,r,\mathrm{Minor}}^{(l)} + (1 - \pi_{e,r}) \cdot \mu_{e,r,\mathrm{Major}}^{(l)} \right), \quad (6)$$

where $\mu_{e,r,\mathrm{Minor}}^{(l)} = \mathbb{E}[X_v^{(l)} \mid v \in \mathcal{N}_r(e), y_v \in \mathrm{Minor}]$ and $\mu_{e,r,\mathrm{Major}}^{(l)} = \mathbb{E}[X_v^{(l)} \mid v \in \mathcal{N}_r(e), y_v \in \mathrm{Major}]$. The minority discriminative signal propagates as follows,

$$\Delta_e^{(l+1)} = \sum_r \pi_{e,r} \cdot W_r^{(l)} \cdot \left( \mu_{e,r,\mathrm{Minor}}^{(l)} - \mu_{e,r,\mathrm{Major}}^{(l)} \right). \quad (7)$$

Taking norms and applying the triangle inequality yields the following formulation,

$$\|\Delta_e^{(l+1)}\| \le \sum_r \pi_{e,r} \cdot \|W_r^{(l)}\| \cdot \|\Delta_e^{(l)}\|. \quad (8)$$

Since $\pi_{e,r} \ll 1$ for minority entities and $\|W_r^{(l)}\|$ is bounded, the update defines a contraction mapping on $\Delta_u^{(l)}$. Therefore, iterating across layers leads to the exponential decay of minority discriminative signal, indicating that the representations of minority and majority entities are indistinguishable.

To mitigate this issue, we propose the *relation-wise gating controller* (Rel-Gate), which aims to enhance the minority-discriminative signal. For each relation type $r$, Rel-Gate first estimates the likelihood of current neighborhood information leaning towards the minority class. A high minority-leaning likelihood implies that the current neighborhood information effectively captures minority predictive patterns which should be enhanced, and the opposite holds when this likelihood is low. Given the neighborhood information of entity $e$ from relation $r$, i.e., message $H_{e,r}$, the relation-wise gating factor is formulated as follows,

$$\Psi_{e,r} = \psi\Bigg( R_r \\ + \mathrm{Softmax}\left( \frac{Q(X_e)K(H_{e,r})^\top}{\sqrt{d}} \right) V(H_{e,r}) \Bigg), \quad (9)$$

where $\psi(\cdot)$ denotes a bounded likelihood estimator function (e.g., Sigmoid), $R_r$ is a learnable embedding corresponding to the relation $r$ (Wang et al., 2025), and $Q(\cdot), K(\cdot), V(\cdot)$ are linear projections corresponding to the query, key, and value transformations (Qiu et al., 2025), respectively. Subsequently, Rel-Gate aggregates the neighborhood messages of diverse relation types according to the estimated gating factor, which can be formulated as follows,

$$X_e \leftarrow \sigma\left( W_e X_e + \sum_r \Psi_{e,r} \odot H_{e,r} \right). \quad (10)$$

Based on the relation-wise gating controller, Rel-MOSS is able to modulate the proportion of minority discriminative information and mitigate the indistinguishable problem of entity representations.

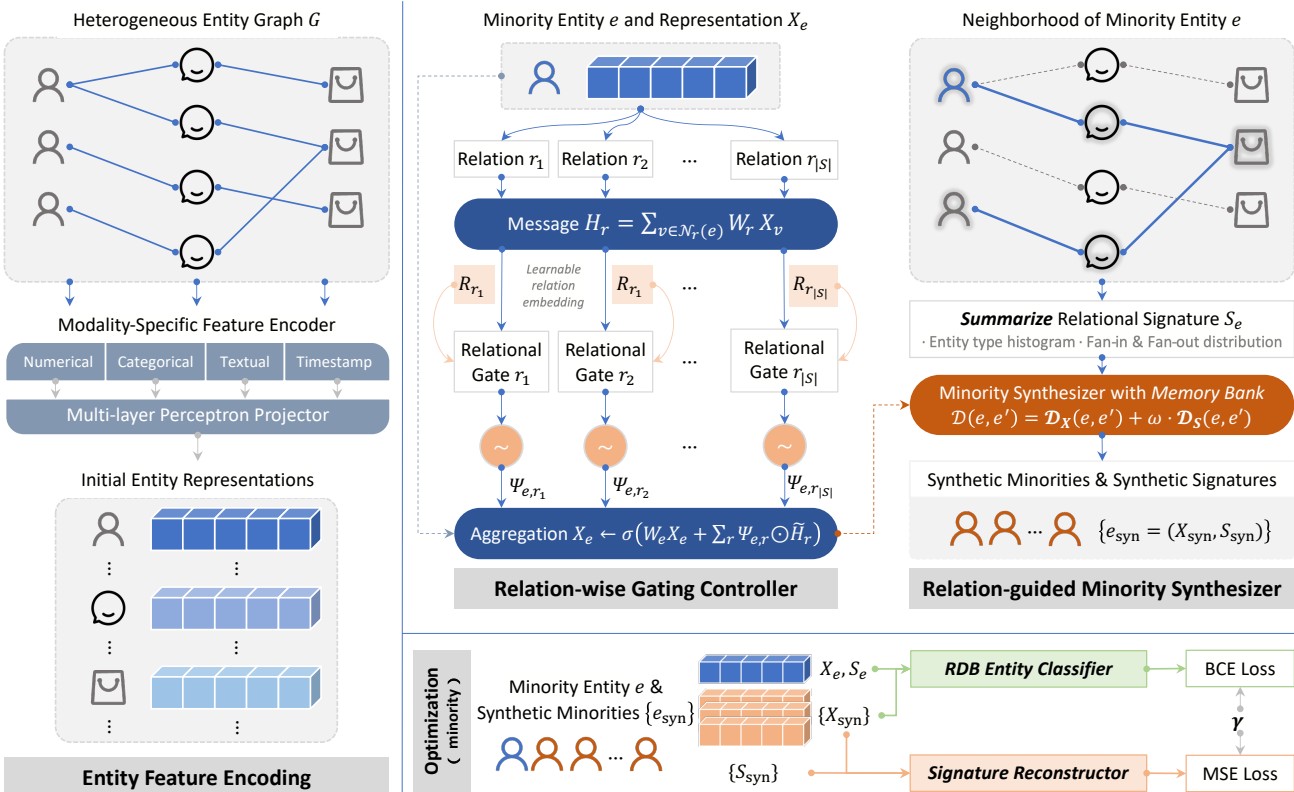

*Figure 3.* Overview of Rel-MOSS. Beginning with the *entity feature encoding*, the original entity features with diverse modalities is converted into unified representations by modality-specific feature encoder. For entity $e$, the *relation-wise gating controller* modulates the neighborhood message from each relation according to the gating factors $\{\Psi_{e,r}\}$. Subsequently, based on the gated representations, the *relation-guided minority synthesizer* integrates entity relational signatures into over-sampling process. Finally, the *optimization* of Rel-MOSS consists of the entity classification and relational signature reconstruction.

## 4.3. Rel-Syn: Relation-guided Minority Synthesizer

Although minority over-sampling techniques serve as the most popular and effective approaches to the class imbalance problem, virtually all the previous methods (Chawla et al., 2002; Zhao et al., 2021; Wu et al., 2022; Li et al., 2023; 2024) focus on homogeneous graphs with relatively simple connectivity patterns. Directly applying previous methods to heterogeneous entity graphs induced from RDBs risks unfaithful sample synthesis, which may disrupt the *relational consistency* and result in sub-optimal performance.

> **Proposition 4.2** (Relational Consistency). *In relational entity graphs, minority entity over-sampling without structural constraints induces a distribution shift in relational roles, which may harm downstream entity classification. Conversely, a synthesis process that preserves relational signatures yields structurally faithful minority over-sampling.*

In RDBs, entity labels are primarily determined by their structural roles rather than by the attached features (Chen et al., 2025). Let $S_e$ denote the structural statistics of an

entity $e$, serving as a signature that identifies its local structural role in the heterogeneous entity graph. Unconstrained interpolation in latent space (Chawla et al., 2002; Zhao et al., 2021) approximates sampling from $\mathbb{P}(X_e \mid y_e \in \text{Minor})$, while ignoring the relational signature distribution $\mathbb{P}(S_e \mid y_e \in \text{Minor})$. However, within the heterogeneous entity graph, the representations and the relational signatures do not conform to isomorphism. As a result, the synthesized entities generated by unconstrained interpolation may exhibit structural roles that deviate from those of true minority entities, introducing structural confounding bias.

In response to this issue, we design the *relation-guided minority synthesizer* (Rel-Syn), which integrates the relational signature $S_e$ of the entity into the synthetic over-sampling process. Given the target entity $e$ belonging to the minor class, Rel-Syn first scrutinizes the structural characteristics within the entity neighborhood, which encapsulate the structural role of the target entity. In our implementation, the adopted structural characteristics include *(i)* the entity type histogram of 1-hop and 2-hop neighbors, and *(ii)* the fan-in and fan-out distribution of relation types, which are empirically beneficial for minority synthesis. Rel-Syn curates the

structural characteristics as the relational signature $S_e$ of the target entity. Along with the gated representation $X_e$ in Section 4.2, the relational signature $S_e$ is passed through the minority synthesis process.

Based on the gated representation $X_e$ and the relational signature $S_e$ of the target entity, Rel-Syn identifies the closest minority sample $e^*$ according to the distance metric below,

$$\mathcal{D}(e, e') = \|X_e - X_{e'}\|_2^2 + \omega \cdot \|S_e - S_{e'}\|_2^2, \qquad (11)$$

where $\omega$ is the weighted hyper-parameter of relational signature constraint. Rel-Syn explicitly conditions on relational signatures to approximate the posterior $\mathbb{P}(X_e \mid S_e, y_e \in \text{Minor})$, thereby preserving relational consistency. To this end, Rel-Syn maintains a memory bank that contains the representations of minorities and their relational signatures. Afterwards, Rel-Syn generates the synthetic minority sample by performing interpolation between $e$ and $e^*$. Additionally, Rel-Syn generates a relational signature for the synthetic minority sample as well, in order to enhance relational consistency. Formally, the synthesis procedure of sample $e_{\text{syn}} = (X_{\text{syn}}, S_{\text{syn}})$ is represented as follows,

$$\begin{aligned} X_{\text{syn}} &= \lambda \cdot X_e + (1 - \lambda) \cdot X_{e^*}, \\ S_{\text{syn}} &= \lambda \cdot S_e + (1 - \lambda) \cdot S_{e^*}, \end{aligned} \qquad (12)$$

where $\lambda$ is the interpolation factor conforming to a $\mathrm{Beta}$ distribution. The label $y_{\text{syn}}$ of the synthetic sample $e_{\text{syn}}$ is assigned to the minority class.

### 4.4. Optimization Objective of Rel-MOSS

The optimization objective of Rel-MOSS is consist of two components: entity classification and relational signature reconstruction. For the original samples (no matter majority or minority) and the synthetic minority samples, Rel-MOSS adopts the binary cross-entropy (BCE) loss as the objective function for entity classification. In addition, for the original minorities and synthetic minorities, in order to facilitate the utilization of relation structural information, Rel-MOSS introduce the mean squared error (MSE) loss as the objective for relational signature reconstruction. The calculation of Rel-MOSS objective can be formulated as follows,

$$\mathcal{L}_{\text{Rel-MOSS}} = \mathcal{L}_{\text{CLS}} + \gamma \cdot \mathcal{L}_{\text{Syn}}, \qquad (13)$$

$$\mathcal{L}_{\text{CLS}} = \sum_e \text{BCE}\Big(f_{\text{CLS}}(\tilde{X}_e), y_e\Big), \qquad (14)$$

$$\mathcal{L}_{\text{Syn}} = \sum_{e \in \text{Minor}} \text{MSE}\Big(f_{\text{Syn}}(\tilde{X}_e), S_e\Big), \qquad (15)$$

$$\tilde{X}_e = \phi(W_x X_e + W_s S_e), \qquad (16)$$

where $\gamma$ is the weighted hyper-parameter. $f_{\text{CLS}}$ and $f_{\text{Syn}}$ denote the classification head and the reconstruction head,

respectively. $\phi(\cdot)$ is a simple MLP to fuse entity representations and relational signatures. As the optimization iterates, Rel-MOSS also refreshes the representations and relational signatures of minorities stored in the memory bank, with first-in-first-out (FIFO) strategy.

## 5. Experiment

To comprehensively validate the practicality of Rel-MOSS[1], we conduct extensive experiments that are designed to investigate the following research questions.

- **RQ1:** How effective is Rel-MOSS compared to classic methods for the class imbalance problem?

- **RQ2:** Can Rel-MOSS handle normal entity classification datasets where the effect of class imbalance is subtle?

- **RQ3:** How do Rel-Gate and Rel-Syn influence the performance of RDB entity classification, respectively?

- **RQ4:** Can Rel-Gate mitigate minority information collapse, and can Rel-Syn generate faithful samples?

- **RQ5:** Under what configurations can Rel-MOSS deliver significant improvements?

### 5.1. Experimental Setup

**Baseline.** To the best of our knowledge, Rel-MOSS is the first method designed for class imbalanced entity classification on RDBs. Hence, classic methods proposed for class imbalance problem are chosen as the baselines, including SMOTE (Chawla et al., 2002), GraphSMOTE (Zhao et al., 2021), RDL with focal loss (Lin et al., 2017), GraphSHA (Li et al., 2023), and ReVar (Yan et al., 2023b).

**Dataset.** We evaluate Rel-MOSS and baselines on Rel-Bench (Robinson et al., 2024), a public benchmark designed for predictive tasks over RDBs. RelBench covers 12 entity classification datasets, each carefully processed from real-world sources across diverse domains such as e-commerce, social networks, and Q&A platforms.

**Evaluation.** To comprehensively evaluate the model capacity for imbalanced entity classification, we adopt the *Balanced Accuracy* (B-Acc) and the *G-Mean* score as evaluation metrics, following the evaluation pipeline in class imbalanced learning (Ma et al., 2025). Details of the baselines, datasets, and metrics are presented in Appendix.B.

### 5.2. Imbalanced Entity Classification Performance

The evaluation results of Rel-MOSS and the baselines on 12 datasets are presented in Table 1. Accordingly, we can draw the following conclusions.

---

[1]Our code is at https://github.com/Esperanto-mega/RelMOSS

*Table 1.* Evaluation results of Rel-MOSS and baselines on 12 RDB entity classification datasets, in the form of mean±std. For both *Balanced Accuracy* (B-Acc) and *G-Mean*, higher values indicate better performance. Underline marks the best baseline, and **bold** indicates that Rel-MOSS or Rel-MOSS variants outperform the best baseline. *Improvement* is defined as (Rel-MOSS − Best-Baseline)/Best-Baseline.

| Dataset | *f1-driver-top3* | | *f1-driver-dnf* | | *avito-user-clicks* | | *avito-user-visits* | |
|---|---|---|---|---|---|---|---|---|
| Metric | B-Acc | G-Mean | B-Acc | G-Mean | B-Acc | G-Mean | B-Acc | G-Mean |
| RDL | 0.5000 | 0.0000 | $0.6274_{\pm0.0692}$ | $0.5835_{\pm0.0995}$ | 0.5000 | 0.0000 | 0.5000 | 0.0000 |
| RDL-HGT | $0.5124_{\pm0.0175}$ | $0.1179_{\pm0.1670}$ | $0.5497_{\pm0.0324}$ | $0.3647_{\pm0.1116}$ | 0.5000 | 0.0000 | 0.5000 | 0.0000 |
| RelGNN | $0.4976_{\pm0.0034}$ | $0.0927_{\pm0.1311}$ | $0.5272_{\pm0.0066}$ | $0.2973_{\pm0.0332}$ | 0.5000 | 0.0000 | 0.5000 | 0.0000 |
| RDL-Focal | 0.5000 | 0.0000 | $0.5871_{\pm0.0094}$ | $0.4940_{\pm0.0485}$ | $0.5491_{\pm0.0080}$ | $0.3465_{\pm0.0294}$ | $0.6086_{\pm0.0098}$ | $0.5376_{\pm0.0265}$ |
| SMOTE | $0.7509_{\pm0.0307}$ | $0.7349_{\pm0.0242}$ | $0.6304_{\pm0.0321}$ | $0.5857_{\pm0.0636}$ | $0.5984_{\pm0.0142}$ | $0.5105_{\pm0.0415}$ | $\underline{0.6255}_{\pm0.0063}$ | $\underline{0.5947}_{\pm0.0216}$ |
| GraphSMOTE | $\underline{0.7521}_{\pm0.0197}$ | $\underline{0.7426}_{\pm0.0243}$ | $0.6135_{\pm0.0211}$ | $0.5724_{\pm0.0305}$ | $\underline{0.6062}_{\pm0.0126}$ | $\underline{0.5335}_{\pm0.0352}$ | $0.6254_{\pm0.0018}$ | $0.5938_{\pm0.0093}$ |
| GraphSHA | $0.5334_{\pm0.0614}$ | $0.2836_{\pm0.1893}$ | $\underline{0.6442}_{\pm0.0305}$ | $\underline{0.6178}_{\pm0.0564}$ | 0.5000 | 0.0000 | $0.5002_{\pm0.0001}$ | $0.0272_{\pm0.0128}$ |
| ReVar | $0.5579_{\pm0.0533}$ | $0.3931_{\pm0.1746}$ | $0.5975_{\pm0.0026}$ | $0.5101_{\pm0.0085}$ | 0.5000 | 0.0000 | $0.5004_{\pm0.0005}$ | $0.0215_{\pm0.0215}$ |
| Rel-MOSS | $\mathbf{0.8098}_{\pm0.0104}$ | $\mathbf{0.8014}_{\pm0.0142}$ | $\mathbf{0.6510}_{\pm0.0221}$ | $\mathbf{0.6262}_{\pm0.0300}$ | $\mathbf{0.6221}_{\pm0.0041}$ | $\mathbf{0.5870}_{\pm0.0041}$ | $\mathbf{0.6298}_{\pm0.0012}$ | $\mathbf{0.6168}_{\pm0.0016}$ |
| w/o Rel-Gate | $\mathbf{0.7651}_{\pm0.0326}$ | $\mathbf{0.7550}_{\pm0.0331}$ | $\mathbf{0.6623}_{\pm0.0158}$ | $\mathbf{0.6454}_{\pm0.0227}$ | $\mathbf{0.6081}_{\pm0.0138}$ | $\mathbf{0.5349}_{\pm0.0380}$ | $0.6252_{\pm0.0019}$ | $0.5947_{\pm0.0094}$ |
| w/o Rel-Syn | $0.5627_{\pm0.0009}$ | $0.1288_{\pm0.0296}$ | $0.6207_{\pm0.0519}$ | $0.5463_{\pm0.1097}$ | 0.5000 | 0.0000 | 0.5000 | 0.0000 |
| *Improvement* | 7.67% | 7.92% | 1.06% | 1.36% | 2.62% | 10.03% | 0.69% | 3.72% |

| Dataset | *event-user-repeat* | | *event-user-ignore* | | *stack-user-engagement* | | *stack-user-badge* | |
|---|---|---|---|---|---|---|---|---|
| Metric | B-Acc | G-Mean | B-Acc | G-Mean | B-Acc | G-Mean | B-Acc | G-Mean |
| RDL | $0.6627_{\pm0.0279}$ | $0.6377_{\pm0.0631}$ | $0.6767_{\pm0.0124}$ | $0.6136_{\pm0.0226}$ | $0.5701_{\pm0.0048}$ | $0.3753_{\pm0.0133}$ | $0.5478_{\pm0.0024}$ | $0.3104_{\pm0.0080}$ |
| RDL-HGT | $0.6020_{\pm0.0737}$ | $0.4296_{\pm0.3045}$ | $0.6820_{\pm0.0059}$ | $0.6235_{\pm0.0147}$ | $0.5489_{\pm0.0007}$ | $0.3142_{\pm0.0022}$ | $0.5139_{\pm0.0087}$ | $0.1600_{\pm0.0510}$ |
| RelGNN | $0.6279_{\pm0.0671}$ | $0.5822_{\pm0.1229}$ | $0.6303_{\pm0.0418}$ | $0.5233_{\pm0.0869}$ | $0.5719_{\pm0.0106}$ | $0.3796_{\pm0.0289}$ | $0.5298_{\pm0.0062}$ | $0.2438_{\pm0.0261}$ |
| RDL-Focal | $0.6145_{\pm0.0602}$ | $0.4865_{\pm0.1800}$ | $0.6905_{\pm0.0025}$ | $0.6407_{\pm0.0054}$ | $0.7050_{\pm0.0194}$ | $0.6484_{\pm0.0324}$ | $0.7042_{\pm0.0028}$ | $0.6528_{\pm0.0046}$ |
| SMOTE | $\underline{0.7108}_{\pm0.0215}$ | $\underline{0.7080}_{\pm0.0218}$ | $0.7151_{\pm0.0069}$ | $0.6898_{\pm0.0160}$ | $0.7903_{\pm0.0063}$ | $0.7780_{\pm0.0090}$ | $0.7676_{\pm0.0038}$ | $0.7500_{\pm0.0055}$ |
| GraphSMOTE | $0.6771_{\pm0.0309}$ | $0.6626_{\pm0.0456}$ | $\underline{0.7269}_{\pm0.0118}$ | $\underline{0.7114}_{\pm0.0189}$ | $\underline{0.7911}_{\pm0.0084}$ | $\underline{0.7815}_{\pm0.0117}$ | $\underline{0.7726}_{\pm0.0030}$ | $\underline{0.7608}_{\pm0.0040}$ |
| GraphSHA | $0.6903_{\pm0.0166}$ | $0.6841_{\pm0.0140}$ | $0.6858_{\pm0.0019}$ | $0.6281_{\pm0.0032}$ | $0.5746_{\pm0.0009}$ | $0.3877_{\pm0.0024}$ | $0.5444_{\pm0.0089}$ | $0.2977_{\pm0.0316}$ |
| ReVar | $0.7002_{\pm0.0121}$ | $0.6964_{\pm0.0095}$ | $0.6893_{\pm0.0057}$ | $0.6366_{\pm0.0106}$ | $0.5851_{\pm0.0108}$ | $0.4134_{\pm0.0273}$ | $0.5480_{\pm0.0049}$ | $0.3106_{\pm0.0160}$ |
| Rel-MOSS | $\mathbf{0.7411}_{\pm0.0072}$ | $\mathbf{0.7399}_{\pm0.0059}$ | $\mathbf{0.7460}_{\pm0.0071}$ | $\mathbf{0.7362}_{\pm0.0103}$ | $\mathbf{0.8112}_{\pm0.0093}$ | $\mathbf{0.8054}_{\pm0.0127}$ | $\mathbf{0.7934}_{\pm0.0021}$ | $\mathbf{0.7861}_{\pm0.0032}$ |
| w/o Rel-Gate | $\mathbf{0.7237}_{\pm0.0132}$ | $\mathbf{0.7225}_{\pm0.0125}$ | $\mathbf{0.7421}_{\pm0.0104}$ | $\mathbf{0.7308}_{\pm0.0138}$ | $0.7903_{\pm0.0079}$ | $0.7777_{\pm0.0109}$ | $\mathbf{0.7802}_{\pm0.0084}$ | $\mathbf{0.7673}_{\pm0.0117}$ |
| w/o Rel-Syn | $0.7108_{\pm0.0253}$ | $0.7051_{\pm0.0213}$ | $0.6913_{\pm0.0054}$ | $0.6380_{\pm0.0082}$ | $0.5745_{\pm0.0086}$ | $0.3868_{\pm0.0222}$ | $0.5473_{\pm0.0032}$ | $0.3085_{\pm0.0106}$ |
| *Improvement* | 4.26% | 4.50% | 2.63% | 3.37% | 2.54% | 3.06% | 2.69% | 3.32% |

| Dataset | *amazon-user-churn* | | *amazon-item-churn* | | *trial-study-outcome* | | *hm-user-churn* | |
|---|---|---|---|---|---|---|---|---|
| Metric | B-Acc | G-Mean | B-Acc | G-Mean | B-Acc | G-Mean | B-Acc | G-Mean |
| RDL | $\underline{0.6309}_{\pm0.0003}$ | $\underline{0.6291}_{\pm0.0005}$ | $0.5194_{\pm0.0093}$ | $0.2002_{\pm0.0576}$ | $0.6127_{\pm0.0050}$ | $0.5933_{\pm0.0038}$ | 0.5000 | 0.0000 |
| RDL-HGT | $0.6106_{\pm0.0083}$ | $0.6072_{\pm0.0106}$ | $0.5990_{\pm0.0301}$ | $0.4923_{\pm0.0716}$ | $0.6074_{\pm0.0180}$ | $0.5870_{\pm0.0402}$ | $0.5423_{\pm0.0182}$ | $0.3347_{\pm0.0745}$ |
| RelGNN | $0.6237_{\pm0.0030}$ | $0.6156_{\pm0.0064}$ | $0.6298_{\pm0.0215}$ | $0.5718_{\pm0.0401}$ | $\underline{0.6242}_{\pm0.0131}$ | $\underline{0.6110}_{\pm0.0277}$ | $\underline{0.5699}_{\pm0.0219}$ | $\underline{0.4384}_{\pm0.0866}$ |
| RDL-Focal | $0.5025_{\pm0.0009}$ | $0.0763_{\pm0.0149}$ | $0.5710_{\pm0.0089}$ | $0.4002_{\pm0.0259}$ | $0.5566_{\pm0.0019}$ | $0.4550_{\pm0.0053}$ | $0.5004_{\pm0.0002}$ | $0.0279_{\pm0.0122}$ |
| SMOTE | $0.6158_{\pm0.0037}$ | $0.5860_{\pm0.0090}$ | $0.6856_{\pm0.0055}$ | $0.6440_{\pm0.0092}$ | $0.5999_{\pm0.0106}$ | $0.5368_{\pm0.0205}$ | $0.5504_{\pm0.0162}$ | $0.3646_{\pm0.0646}$ |
| GraphSMOTE | $0.6099_{\pm0.0059}$ | $0.5709_{\pm0.0166}$ | $\underline{0.7001}_{\pm0.0042}$ | $\underline{0.6768}_{\pm0.0067}$ | $0.5979_{\pm0.0183}$ | $0.5440_{\pm0.0537}$ | $0.5557_{\pm0.0040}$ | $0.3898_{\pm0.0146}$ |
| GraphSHA | $0.6347_{\pm0.0014}$ | $0.6277_{\pm0.0043}$ | $0.6721_{\pm0.0159}$ | $0.6282_{\pm0.0296}$ | $0.6193_{\pm0.0036}$ | $0.5959_{\pm0.0051}$ | $0.5628_{\pm0.0012}$ | $0.4288_{\pm0.0016}$ |
| ReVar | $0.6311_{\pm0.0017}$ | $0.6298_{\pm0.0002}$ | $0.6455_{\pm0.0319}$ | $0.5737_{\pm0.0645}$ | $0.6004_{\pm0.0144}$ | $0.5946_{\pm0.0146}$ | $0.5618_{\pm0.0026}$ | $0.4271_{\pm0.0070}$ |
| Rel-MOSS | $\mathbf{0.6363}_{\pm0.0015}$ | $\mathbf{0.6328}_{\pm0.0032}$ | $\mathbf{0.7108}_{\pm0.0055}$ | $\mathbf{0.6942}_{\pm0.0089}$ | $\mathbf{0.6252}_{\pm0.0030}$ | $0.6023_{\pm0.0038}$ | $\mathbf{0.5736}_{\pm0.0024}$ | $\mathbf{0.4537}_{\pm0.0036}$ |
| w/o Rel-Gate | $0.6043_{\pm0.0023}$ | $0.5556_{\pm0.0063}$ | $\mathbf{0.7036}_{\pm0.0041}$ | $\mathbf{0.6887}_{\pm0.0066}$ | $\mathbf{0.6273}_{\pm0.0019}$ | $\mathbf{0.6126}_{\pm0.0028}$ | $0.5616_{\pm0.0053}$ | $0.4173_{\pm0.0137}$ |
| w/o Rel-Syn | $\mathbf{0.6455}_{\pm0.0003}$ | $\mathbf{0.6318}_{\pm0.0009}$ | $0.6908_{\pm0.0023}$ | $0.6608_{\pm0.0039}$ | $0.6151_{\pm0.0027}$ | $0.6028_{\pm0.0002}$ | $\mathbf{0.5956}_{\pm0.0062}$ | $\mathbf{0.4969}_{\pm0.0188}$ |
| *Improvement* | 0.86% | 0.59% | 1.53% | 2.57% | 0.16% | -1.42% | 0.65% | 3.49% |

***First*, Rel-MOSS can effectively mitigate the class imbalance problem in RDB entity classification tasks (RQ1).** According to Table 1, one can notice that for some datasets, including `f1-driver-top3`, `avito-user-clicks`, and so on, the B-Acc and G-Mean metrics of native RDL models (RDL, RDL-HGT, and RelGNN) approach 0.5 and 0.0, re- spectively. Revisiting their definitions in formulas 17 and 18, this phenomenon implies that these datasets suffer from a severe class imbalance, and the classifier fails to distinguish between the positives and the negatives. Compared with classic methods targeting the class imbalance problem (Fo- cal loss, SMOTE, and GraphSMOTE), Rel-MOSS exhibits

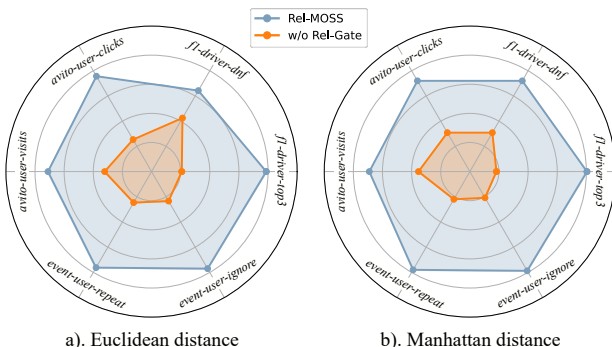

Figure 4. Comparison between Rel-MOSS and w/o Rel-Gate in terms of a). Euclidean distance, and b). Manhattan distance.

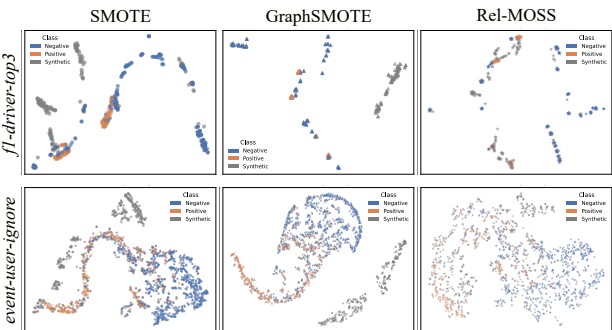

Figure 5. Visualization of entity representations learned by SMOTE, GraphSMOTE, and Rel-MOSS.

significant superiority and achieves an average improvement of 3.22% and 6.49% in terms of B-Acc and G-Mean.

***Second*, Rel-MOSS can maintain the classification capability on normal datasets unaltered, by generating faithful synthetic samples (RQ2).** Aside from the datasets suffering from class imbalance, some other datasets, especially `f1-driver-dnf`, `amazon-user-churn`, and `trial-study-outcome`, can be classified normally by native RDL models with acceptable performance. Meanwhile, as presented by Table 1, integrating classic techniques for class imbalance conversely degrades classification performance. The degradation could be traced to the synthetic over-sampling process, where the relational consistency is ignored. In contrast, Rel-MOSS remains capable of achieving the best or top-tier entity classification capability, which owes to the faithful synthesis ability of Rel-Syn .

***Third*, for imbalanced RDB entity classification, the relation-guided minority synthesizer acts as the fundamental determinant, while the relation-wise gating controller can further enhance the entity classification capability (RQ3).** By investigating the Rel-MOSS variants *w/o Rel-Gate* and *w/o Rel-Syn*, one can notice that in most scenarios, removing Rel-Gate incurs a slight performance drop compared to removing Rel-Syn. On the one hand, the considerable effectiveness of Rel-Syn aligns with the consensus in previous research that appropriate synthetic minority oversampling might be the most effective approach for addressing class imbalance. On the other hand, the gap between Rel-MOSS and w/o Rel-Gate demonstrates that the relation-wise gating controller is able to mitigate the minority information collapse and thus improve the overall classification performance.

### 5.3. In-depth Qualitative Analysis (RQ4)

**Relation-wise Gating Controller.** To assess how well Rel-Gate enhances the distinctiveness of entity representations, we report the average distance between the representation

centroids of minority entities and those of majority entities on the test sets. The results, in terms of Euclidean and Manhattan distances, are presented in Figure 4. The Euclidean distance is influenced primarily by whether the largest coordinate difference is aligned, whereas the Manhattan distance captures how many coordinates differ. According to the radar chart, it can be observed that the representation distances of Rel-MOSS are consistently larger than those of w/o Rel-Gate across all evaluated datasets. This phenomenon suggests that Rel-Gate is able to improve the distinguishability of the entity representations.

**Relation-guided Minority Synthesizer.** In order to comprehensively investigate the synthesis quality of Rel-Syn, as shown in Figure 5, we visualize the synthetic entities via the $t$-SNE algorithm and introduce the entities generated by SMOTE and GraphSMOTE for comparison. Specifically, the evaluated datasets, `f1-driver-top3` and `event-user-ignore`, both suffer from rare positive entities, with imbalance ratios of 4.86 and 4.93, respectively. For SMOTE and GraphSMOTE, the distribution of synthetic minorities, denoted by gray points, severely diverges from the true minorities distribution denoted by orange points. In contrast, Rel-MOSS is able to generate faithful minority entities that conform to the true minority manifold, owing to the guidance of the relational signature.

### 5.4. Hyper-parameter Investigation (RQ5)

For a comprehensive understanding of the Rel-MOSS characteristics, we conduct an extensive investigation of the model hyper-parameters, including *(i)* the weighted parameter $\gamma$ in Rel-MOSS loss defined by Eq.13, *(ii)* the weighted parameter $\omega$ in Rel-Syn distance metric defined by Eq.11, and *(iii)* the learning rate $\eta$. The complete results are presented in Appendix.C, and the following conclusions can be drawn. ***First*, the relational signature reconstruction objective can deliver consistent performance gains. According to Figure 6, Rel-MOSS with $\gamma$ ranging from 0.1 to 2 yields stable improvements over the zero-$\gamma$ variant. Particularly, Rel-MOSS achieves an average improvement of up to 9.71%

*Table 2.* Evaluation results comparison in terms of PR-AUC ↑ and ROC-AUC ↑.

| Dataset | f1-driver-top3 | | f1-driver-dnf | | event-user-repeat | | event-user-ignore | |
|---|---|---|---|---|---|---|---|---|
| Metric | PR-AUC | ROC-AUC | PR-AUC | ROC-AUC | PR-AUC | ROC-AUC | PR-AUC | ROC-AUC |
| RDL | $0.2796_{\pm0.0035}$ | $0.7572_{\pm0.0112}$ | $0.2030_{\pm0.0044}$ | $\underline{0.7091_{\pm0.0094}}$ | $0.7012_{\pm0.0103}$ | $0.7674_{\pm0.0102}$ | $0.4975_{\pm0.0087}$ | $0.8092_{\pm0.0088}$ |
| RelGNN | $\underline{0.3085_{\pm0.0149}}$ | $\underline{0.7792_{\pm0.0105}}$ | $\underline{0.2081_{\pm0.0100}}$ | $0.6911_{\pm0.0123}$ | $0.6323_{\pm0.0106}$ | $0.7212_{\pm0.0145}$ | $0.4525_{\pm0.0069}$ | $0.8301_{\pm0.0075}$ |
| SMOTE | $0.2993_{\pm0.0250}$ | $0.7369_{\pm0.0185}$ | $0.1996_{\pm0.0057}$ | $0.7017_{\pm0.0087}$ | $\underline{0.7155_{\pm0.0099}}$ | $\underline{0.7808_{\pm0.0091}}$ | $\underline{0.5028_{\pm0.0161}}$ | $\underline{0.8318_{\pm0.0114}}$ |
| Rel-MOSS | $\mathbf{0.3571_{\pm0.0071}}$ | $\mathbf{0.7999_{\pm0.0058}}$ | $\mathbf{0.2372_{\pm0.0018}}$ | $\mathbf{0.7153_{\pm0.0042}}$ | $\mathbf{0.7419_{\pm0.0024}}$ | $\mathbf{0.7959_{\pm0.0063}}$ | $\mathbf{0.5308_{\pm0.0027}}$ | $\mathbf{0.8524_{\pm0.0055}}$ |

and 18.07% regarding balanced accuracy and G-Mean on `f1-driver-top3` dataset. On the other hand, the gains diminish or even worsen when $\gamma$ becomes abnormally large (e.g., 5 or 10). **Second**, relatively large values of $\omega$ are suitable for most datasets. As illustrated in Figure 7, when $\omega$ exceeds 10, the improvements over the zero-$\omega$ variant become substantial for most datasets (4 out of 6). In contrast, a relatively small $\omega$ tends to deteriorate performance. **Third**, relatively small values of the learning rate $\eta$ are beneficial for optimization. In Figure 8, for most datasets, a learning rate smaller than 5e-3 can deliver optimal performance.

### 5.5. Ranking-based Evaluation

To investigate the ranking performance of Rel-MOSS, this section compares it with the baselines in terms of the PR-AUC and ROC-AUC metrics. As shown in Table 2, the evaluated result demonstrates that Rel-MOSS consistently and significantly outperforms all competitors across all evaluated datasets on both PR-AUC and ROC-AUC. The superiority is particularly pronounced in PR-AUC, which is widely recognized as a more informative and sensitive metric for highly imbalanced classification tasks. In detail, Rel-MOSS improves the PR-AUC from 0.3085 of RelGNN to 0.3571 on the `f1-driver-top3` dataset, and from 0.5028 of SMOTE to 0.5308 on the `event-user-ignore` dataset.

In summary, Rel-MOSS can not only boost the holistic ranking capacity but also achieve better classification performance under the default threshold.

### 5.6. Time Efficiency

To investigate the computational cost of Rel-MOSS compared with the standard RDL pipeline, we present a theoretical analysis of the additional operation of Rel-MOSS and a practical runtime comparison in Table .

• **Theoretical Analysis**. Based on the standard RDL pipeline, Rel-MOSS introduces a gating mechanism formulated in Eq.9 and a search process according to the distance defined in Eq.11. Specifically, the gating mechanism performs a cross attention operation. Let $B$ and $d$ denote the batch size and the embedding dimension, the time complexity of the cross attention operation is $O(B^2d)$. For all the $|R|$ relations, the total time complexity induced by Rel-

*Table 3.* Practical runtime (s) comparison of RDL and Rel-MOSS.

| Dataset | f1-driver-top3 | f1-driver-dnf | event-user-repeat | event-user-ignore |
|---|---|---|---|---|
| Standard RDL | 1.0401 | 6.1067 | 2.9947 | 13.0044 |
| Rel-MOSS | 0.9979 | 5.7454 | 3.6566 | 14.0246 |
| # Sample | 1353 | 11411 | 3842 | 19239 |
| *Imb-ratio* | 4.86 | 7.36 | 1.04 | 4.92 |

Gate is thus $O(|R| \cdot B^2d)$. Regarding the search process in Rel-Syn, let $U$ denote the scale of the over-sampling memory bank, the time complexity of the distance calculation is $O(UBd)$, and the following sort complexity is $O(U \log U)$. Therefore, the total complexity induced by Rel-Syn is $O(UBd + U \log U)$. In summary, the additional computational cost of Rel-MOSS is linear relative to the number of relation types, linearithmic to the scale of the Rel-Syn memory bank, and quadratic to the batch size.

• **Practical Runtime**. We present the practical runtime in Table 3. The runtime of Rel-MOSS and standard RDL is of the same magnitude. For the `f1-driver-top3` and `f1-driver-dnf` datasets, the runtime of Rel-MOSS is reduced by down-sampling the majority samples. Even on the larger `event-user-ignore` dataset, which contains over 19,000 samples, the additional runtime cost is minimal (approximately 1.0 second increase per epoch). This demonstrates that Rel-MOSS successfully enhances minority classification performance without sacrificing the scalability and practical deployment speed required for real-world relational database applications.

## 6. Conclusion

We investigate, for the first time, the class imbalance problem along with its ramifications in the RDB entity classification task. To overcome this problem, we propose the relation-centric minority synthetic over-sampling GNN (Rel-MOSS), which consists of a relation-wise gating controller (Rel-Gate) and a relation-guided minority synthesizer (Rel-Syn). In particular, Rel-Gate aims to modulate the neighborhood message for each relation type, and Rel-Syn integrates structural statistics into the minority synthesis for relational consistency. Experiments on 12 entity classification datasets demonstrate the superiority of Rel-MOSS over SOTA RDL methods and classic methods for handling class imbalance.

## Acknowledgement

J Yin and H Yan are supported by the Graduate Innovation Project of Central South University (No.1053320252239). In particular, we must acknowledge Mr. Chenhan Wang from Hyper.AI for the generous support of the essential computing resources.

## Impact Statement

This work aims to impact the reliability of relational deep learning in real-world applications, such as e-commerce, social media, and healthcare, where data is inherently imbalanced. By effectively addressing class imbalance, Rel-MOSS enhances the detection of critical, rare events, such as identifying fraudulent accounts or predicting medical trial outcomes, thereby reducing financial losses and improving platform integrity. Ethically, this approach mitigates the algorithmic bias that typically causes models to ignore or under-represent minority entities.

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

## A. Notation

In Table 4, we summarize the main notations used throughout this manuscript.

*Table 4.* Notations and corresponding descriptions.

| Notation | Description |
|---|---|
| $\mathcal{R}$ | Relational database (RDB) |
| $\mathcal{T}$ | Set of tables in the RDB |
| $T, T_t$ | A specific table in the relational database |
| $\mathcal{S}$ | Set of linkages induced from the RDB foreign-primary key reference |
| $e, e_{\text{syn}}$ | RDB entity, synthetic entity |
| $e_m$ | The $m$-th feature of entity $e$ |
| $\mathcal{P}$ | Primary key of RDB |
| $\mathcal{F}$ | Set of RDB foreign keys |
| $f_i \in \mathcal{F}$ | Foreign key of RDB |
| $\mathcal{V}^t$ | Node set collected from the $t$-th RDB table |
| $\mathcal{E}^r$ | Edge set collected from the $r$-th RDB relation |
| $E_k$ | Encoder on feature of modality $k$ |
| $X_e$ | Representation of entity $e$ |
| $\mathcal{N}_r(e)$ | Neighborhood of entity $e$ according to relation $r$ |
| $v \in \mathcal{N}_r(e)$ | Neighbor entity in the neighborhood $\mathcal{N}_r(e)$ |
| $\Delta_e^{(l)}$ | Minority discriminative signal of entity $e$ at the $l$-th message passing |
| $\pi_{e,r}$ | Proportion of minority neighbors of entity $e$ according to relation $r$ |
| $H_{e,r}$ | Neighborhood message of entity $e$ from relation $r$ |
| $\psi(\cdot)$ | Bounded likelihood estimator function |
| $R_r$ | Learnable embedding of relation $r$ |
| $Q(\cdot), K(\cdot), V(\cdot)$ | Linear projections of query, key, and value |
| $S_e$ | Relational signature of entity $e$ |
| $\mathcal{D}(\cdot, \cdot)$ | Distance metric of Rel-Syn |
| $\mathcal{D}_X(\cdot, \cdot)$ | Distance metric based on entity representations |
| $\mathcal{D}_S(\cdot, \cdot)$ | Distance metric based on entity relational signatures |
| $\omega$ | Weight parameter in $\mathcal{D}(\cdot, \cdot) = \mathcal{D}(\cdot, \cdot) + \omega \cdot \mathcal{D}_S(\cdot, \cdot)$ |
| $\lambda$ | Interpolation factor |
| $\gamma$ | Weight parameter in Rel-MOSS loss |

## B. Experimental Detail

In the main experiment, we implement Rel-MOSS based on the standard RDL pipeline, adopting the heterogeneous GraphSAGE as the GNN backbone. The evaluated baselines, datasets, and metrics are introduced in detail as follows.

### B.1. Baseline

- **Standard RDL** (Robinson et al., 2024) begins with a modality-wise encoder based on tabular ResNet (Gorishniy et al., 2021) that projects raw row attributes into initial node states, utilizing learnable embedding lookups for categorical features and linear layers for numerical data. Structural dependencies are then captured via a Heterogeneous GraphSAGE (Hamilton et al., 2017) backbone, which performs relation-specific message passing to aggregate information across foreign-primary connections. Finally, the learned representations of the target nodes are fed into a multi-layer perceptron prediction head to generate the downstream task output.

- **RDL-HGT** (Dwivedi et al., 2025) employs the Heterogeneous Graph Transformer (HGT) (Hu et al., 2020) as a representative Graph Transformer baseline, implemented via the `HGTConv` operator from PyTorch Geometric (Fey & Lenssen, 2025) to replace the default GNN module within the RDL pipeline. The architecture matches the standard RDL configuration with 2 layers, 4 attention heads, residual connections, and layer normalization.

- **Rel-GNN** (Chen et al., 2025) introduces a Composite Message Passing framework designed to address structural inefficiencies (such as bridge and hub nodes) inherent in relational entity graphs. The architecture uses atomic routes that systematically compute paths avoiding intermediate junction tables, allowing direct, single-hop interactions between logically related entities. Specifically, RelGNN employs a fuse operation to linearly combine features from source and intermediate nodes, followed by an aggregate operation using multi-head attention to update the destination node.

- **RDL-Focal** utilizes focal loss (Lin et al., 2017) to mitigate class imbalance. This loss function reshapes the cross-entropy loss to reduce the relative importance of easy negatives, prioritizing the learning of hard, misclassified samples.

- **SMOTE** (Chawla et al., 2002), the acronym for synthetic minority over-sampling technique, augments the dataset by generating synthetic training examples rather than replicating existing ones. The algorithm operates in feature space by performing linear interpolation between a minority class sample and its $k$-nearest neighbors, thereby enriching the minority distribution and reducing the risk of overfitting.

- **GraphSMOTE** (Zhao et al., 2021) addresses the class imbalance problem in semi-supervised node classification by synthesizing minority samples within a latent embedding space learned by a GNN feature extractor, rather than interpolating in the raw high-dimensional feature domain. To integrate these synthetic nodes into the graph topology, the framework simultaneously trains an edge generator to predict connectivity relationships for the newly generated samples. This approach results in an augmented, class-balanced graph structure that allows the downstream GNN classifier to learn more effective representations for under-represented classes. In our experiment, we extend GraphSMOTE from homogeneous graphs to heterogeneous graphs as a comparable baseline.

## B.2. Dataset

As stated in Section 3, we focus on the RDB entity classification tasks, and detailed task descriptions are listed as follows.

- `f1-driver-top3` predicts whether a driver will secure a top-three qualifying position in a race within the next month.

- `f1-driver-dnf` predicts whether a driver will fail to finish a race within the next month.

- `avito-user-clicks` predicts whether each customer will click on more than one advertisement in the next 4 days.

- `avito-user-visits` predicts whether each customer will visit more than one advertisement in the next 4 days.

- `event-user-repeat` predicts whether a user will attend an event (by responding yes or maybe) in the next 7 days if they have already attended an event in the last 14 days.

- `event-user-ignore` predicts whether a user will ignore more than 2 event invitations in the next 7 days

- `stack-user-engagement` predicts whether a user will make any votes, posts, or comments in the next 3 months.

- `stack-user-badge` predicts whether a user will receive a new badge in the next 3 months.

- `amazon-user-churn` predicts 1 if the customer does not review any product in the next 3 months and 0 otherwise.

- `amazon-item-churn` predicts 1 if the product does not receive any reviews in the next 3 months.

- `trial-study-outcome` predicts whether the trials will achieve their primary outcome (defined as $p$-value < 0.05).

- `hm-study-churn` predicts whether a customer with no transactions will churn in the upcoming week.

The dataset statistics are available in the RelBench repository[2] (Robinson et al., 2024). Furthermore, in Table 5, we summarize the extent of imbalance for each dataset.

---

[2]https://relbench.stanford.edu

*Table 5.* Extent of class imbalance for each dataset. *Imb-ratio* represents the imbalance ratio, which is defined as # Majority/# Minority. The statistics is summarized on the train set, which determines the model training dynamics.

| Dataset | *f1-driver-top3* | *f1-driver-dnf* | *avito-user-clicks* | *avito-user-visits* |
|---|---|---|---|---|
| # Positive | 231 | 10046 | 2302 | 78467 |
| # Negative | 1122 | 1365 | 57152 | 8152 |
| *Imb-ratio* | 4.86 | 7.36 | 24.83 | 9.62 |
| **Dataset** | *event-user-repeat* | *event-user-ignore* | *stack-user-engagement* | *stack-user-badge* |
| # Positive | 1882 | 3247 | 68020 | 163048 |
| # Negative | 1960 | 15992 | 1292830 | 3223228 |
| *Imb-ratio* | 1.04 | 4.92 | 19.01 | 19.77 |
| **Dataset** | *amazon-user-churn* | *amazon-item-churn* | *trial-study-outcome* | *hm-user-churn* |
| # Positive | 2956658 | 1113863 | 7647 | 3170367 |
| # Negative | 1775897 | 1445401 | 4347 | 701043 |
| *Imb-ratio* | 1.66 | 1.30 | 1.76 | 4.52 |

## B.3. Metric

Following the evaluation pipeline in class imbalanced learning (Ma et al., 2025), we adopt the *Balanced Accuracy* (B-Acc) and the *G-Mean* as evaluation metrics, which are defined as follows.

$$\text{B-Acc} = \frac{1}{2}\Big(\frac{\text{TP}}{\text{TP} + \text{FN}} + \frac{\text{TN}}{\text{FP} + \text{TN}}\Big), \tag{17}$$

$$\text{G-Mean} = \sqrt{\frac{\text{TP}}{\text{TP} + \text{FN}} \cdot \frac{\text{TN}}{\text{FP} + \text{TN}}}, \tag{18}$$

where TP, TN, FP, FN represents true positives, true negatives, false positives, and false negatives, respectively.

Here, we also provide the definition of the ROC curve and the PR curve. *For the ROC curve*, the $x$-axis and the $y$-axis represent the true positive ratio $\text{TP}/(\text{TP} + \text{FN})$ and the false positive ratio $\text{FP}/(\text{FP} + \text{TN})$, respectively. *For the PR curve*, the $x$-axis and the $y$-axis represent the Precision $\text{TP}/(\text{TP} + \text{FP})$ and the Recall $\text{TP}/(\text{TP} + \text{FN})$, respectively.

## C. Hyper-parameter Analysis

Owing to the layout restrictions of the manuscript, we present the detailed results of hyper-parameter analyzes as follows.

• $\gamma$ **in** $\mathcal{L}_{\text{Rel}-\text{MOSS}}$. As defined in Eq.13, $\gamma$ controls the intensity of the signature reconstruction objective $\mathcal{L}_{\text{Syn}}$. In Figure 6, we illustrate the variation tendency of entity classification performance, where $\gamma$ takes values from the set $\{0, 0.1, 0.5, 1, 2, 5, 10\}$. When $\gamma$ is restricted to $\{0.1, 0.5, 1, 2\}$, the incorporation of $\mathcal{L}_{\text{Syn}}$ can ensure stable improvements. However, $\gamma$ assuming extremely large values, such as 5 and 10, tends to significantly deteriorate classification capacity.

• $\omega$ **in** $\mathcal{D}(\cdot, \cdot)$. As defined in Eq.11, the hyper-parameter $\omega$ controls the intensity of the relational signature distance $\mathcal{D}_S$ during minority over-sampling. In Figure 7, as $\omega$ takes values from $\{0, 1, 5, 10, 50, 100\}$, we present the tendency of variation in entity classification performance. Based on the results, we observe that using relatively large values of $\omega$ is advantageous for most datasets.

• **Learning rate** $\eta$. In Figure 8, we present the entity classification performance as the learning rate $\eta$ grows from 1e-4 to 1e-2. According to the results, we conclude that a relatively small learning rate is beneficial for Rel-MOSS optimization.

• **Latent dimension** $d$. As shown in Table 6, we investigate the impact of the latent embedding dimension $d$. According to the result, we observe that $d = 128$ consistently yields the best performance on three out of the four datasets (f1-driver-top3, event-user-repeat, and event-user-ignore), indicating that this dimension offers sufficient capacity to encode the complex relational dependencies inherent in the data. However, further increasing the dimension to 256 or 512 leads to a sharp performance decline (e.g., a 25% drop in G-Mean on f1-driver-top3 when moving from $d = 128$ to $d = 512$).

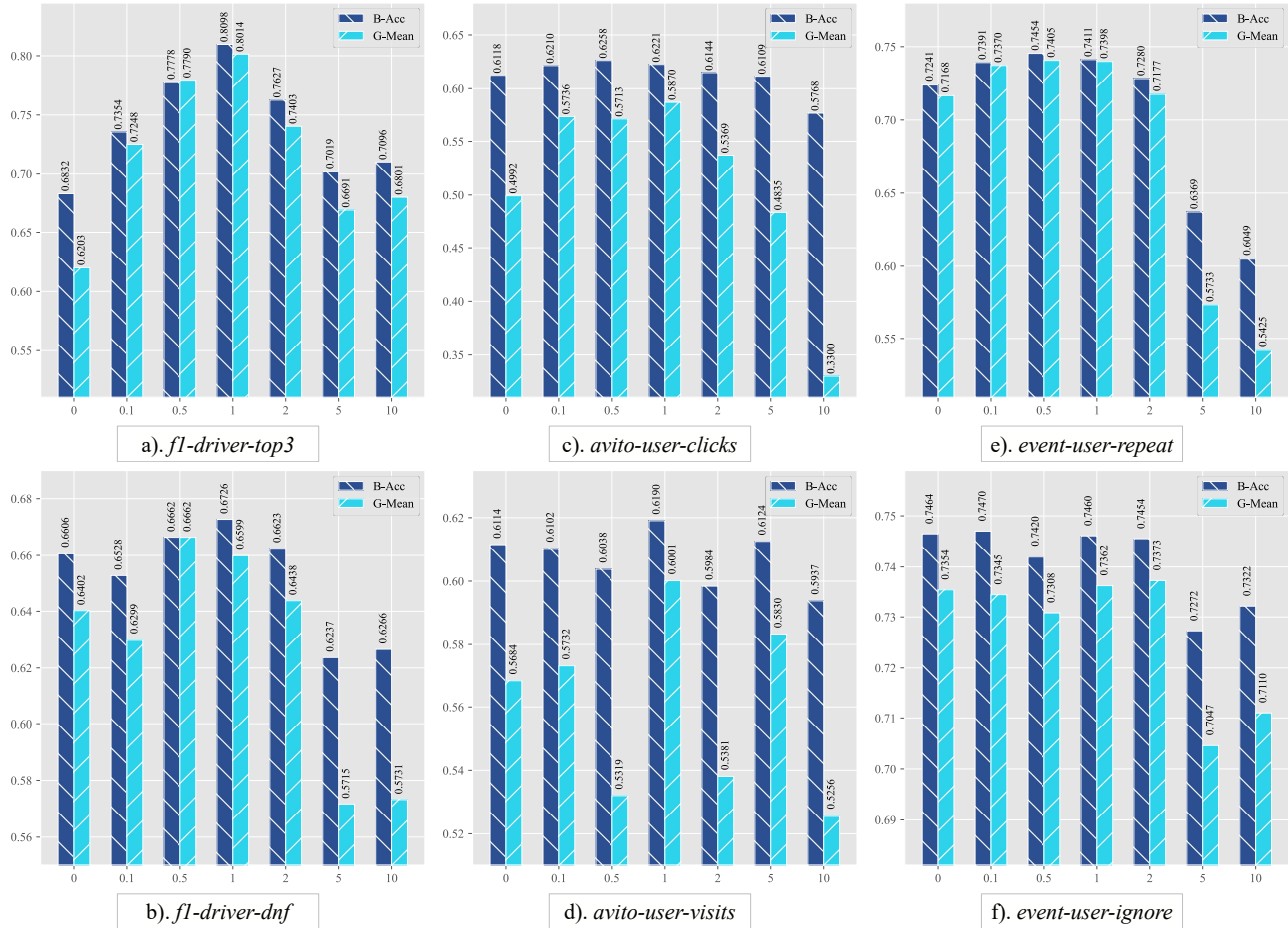

*Figure 6.* Performance of Rel-MOSS versus $\gamma$ in Eq.13 on 6 datasets, in terms of balanced accuracy (B-Acc) and G-Mean.

This phenomenon suggests that excessive model capacity induces overfitting, particularly in class-imbalanced scenarios where the model may overfit to the minority noise.

• **Memory scale** $U$. We further investigate the impact of the memory bank scale $U$ on classification performance, as detailed in Table 7. The memory bank size determines the diversity of historical minority representations available for the Rel-Syn. The empirical results demonstrate that U=1024 serves as an optimal configuration for most datasets, striking a balance between sample diversity and relevance. Specifically, on the f1-driver-top3 and event-user-ignore datasets, the model achieves peak performance at this scale, with G-Mean scores reaching 0.8014 and 0.7362, respectively. In contrast, a smaller memory bank (i.e., $U = 256$) consistently leads to suboptimal performance, likely due to an insufficient number of stored prototypes to accurately span the minority class manifold. Conversely, excessively increasing the scale to $U = 2048$ yields diminishing returns or slight degradation. This suggests that an overly large buffer may retain stale embeddings that are no longer aligned with current training dynamics, thereby introducing noise into the interpolation process.

• **Batch size** $B$. As shown in Table 8, we investigate the impact of the batch size $B$. The results indicate that $B = 512$ generally delivers the most consistent and robust performance across the evaluated datasets. Specifically, for the event-user-repeat and event-user-ignore tasks, the model achieves optimal G-Mean scores of 0.7398 and 0.7362 at $B = 512$. Increasing the batch size further to $B = 1024$ leads to a noticeable degradation in these cases, likely due to a reduction in the stochastic noise that aids in generalization. However, we observe that f1-driver-top3 benefits significantly from larger batch sizes, peaking at $B = 1024$ (i.e., 0.8138 G-Mean), which suggests that certain complex tasks require the more stable gradient estimation provided by larger batches.

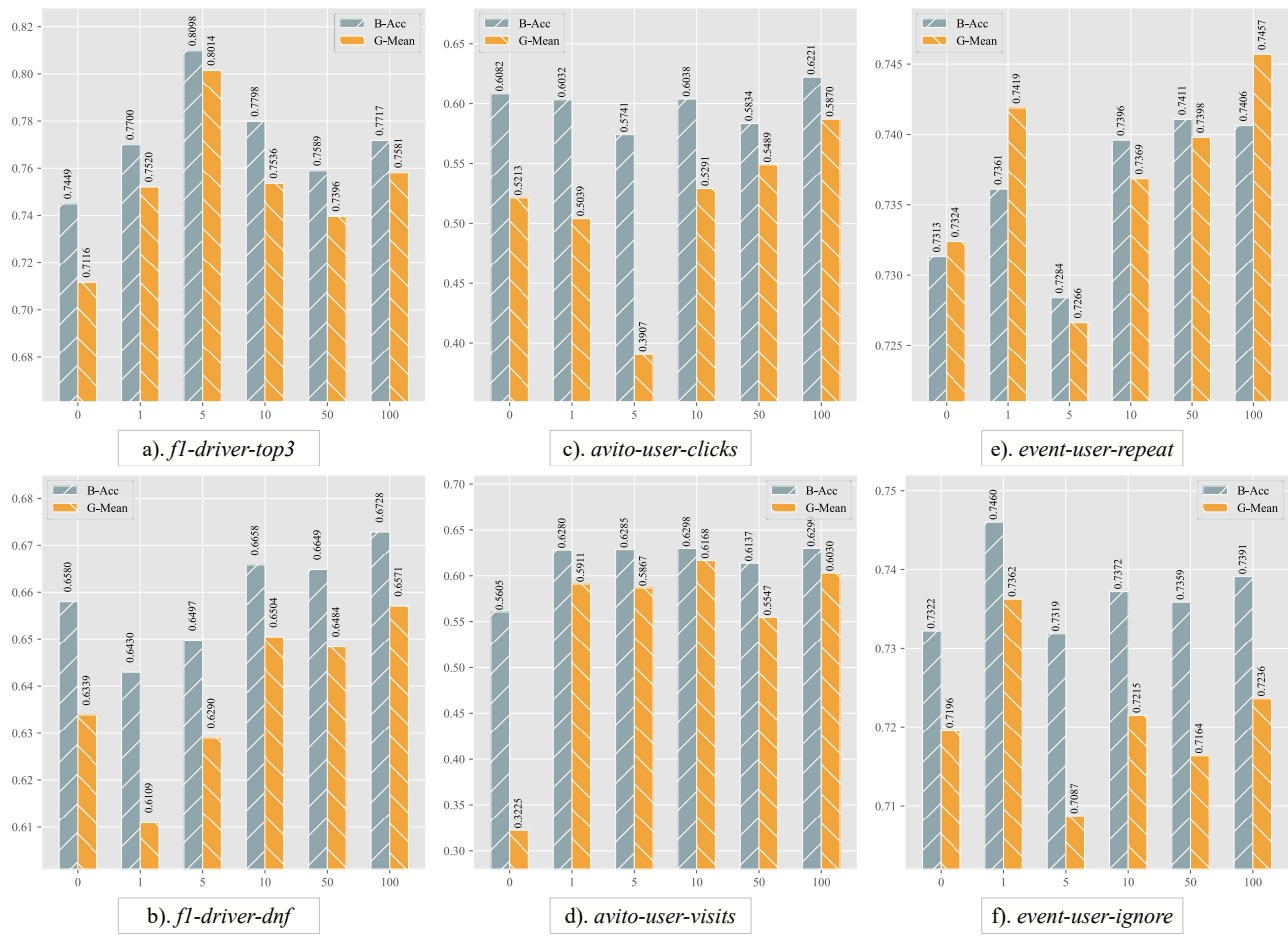

*Figure 7.* Performance of Rel-MOSS versus $\omega$ in Eq.11 on 6 datasets, in terms of balanced accuracy (B-Acc) and G-Mean.

*Table 6.* Entity classification performance of Rel-MOSS versus latent dimension $d$.

| Dataset | *f1-driver-top3* | | *f1-driver-dnf* | | *event-user-repeat* | | *event-user-ignore* | |
|---|---|---|---|---|---|---|---|---|
| Metric | **B-Acc** | **G-Mean** | **B-Acc** | **G-Mean** | **B-Acc** | **G-Mean** | **B-Acc** | **G-Mean** |
| $d = 32$ | $0.7345_{\pm 0.0127}$ | $0.7274_{\pm 0.0146}$ | $0.7184_{\pm 0.0195}$ | $0.7167_{\pm 0.0064}$ | $0.7317_{\pm 0.0246}$ | $0.7276_{\pm 0.0284}$ | $0.7263_{\pm 0.0072}$ | $0.7032_{\pm 0.0131}$ |
| $d = 64$ | $0.7886_{\pm 0.0134}$ | $0.7861_{\pm 0.0121}$ | $0.6717_{\pm 0.0071}$ | $0.6589_{\pm 0.0145}$ | $0.6991_{\pm 0.0273}$ | $0.6981_{\pm 0.0134}$ | $0.7101_{\pm 0.0020}$ | $0.6816_{\pm 0.0058}$ |
| $d = 128$ | $0.8098_{\pm 0.0104}$ | $0.8014_{\pm 0.0142}$ | $0.6510_{\pm 0.0221}$ | $0.6262_{\pm 0.0300}$ | $0.7411_{\pm 0.0072}$ | $0.7398_{\pm 0.0059}$ | $0.7460_{\pm 0.0071}$ | $0.7362_{\pm 0.0103}$ |
| $d = 256$ | $0.7122_{\pm 0.0119}$ | $0.7122_{\pm 0.0164}$ | $0.6755_{\pm 0.0281}$ | $0.6662_{\pm 0.0540}$ | $0.6864_{\pm 0.0249}$ | $0.6677_{\pm 0.0620}$ | $0.7210_{\pm 0.0158}$ | $0.6918_{\pm 0.0284}$ |
| $d = 512$ | $0.6203_{\pm 0.0195}$ | $0.5513_{\pm 0.0607}$ | $0.6830_{\pm 0.0301}$ | $0.6777_{\pm 0.0388}$ | $0.6778_{\pm 0.0216}$ | $0.6722_{\pm 0.0101}$ | $0.6988_{\pm 0.0150}$ | $0.6583_{\pm 0.0298}$ |

*Table 7.* Entity classification performance of Rel-MOSS versus memory bank scale $U$.

| Dataset | *f1-driver-top3* | | *f1-driver-dnf* | | *event-user-repeat* | | *event-user-ignore* | |
|---|---|---|---|---|---|---|---|---|
| Metric | **B-Acc** | **G-Mean** | **B-Acc** | **G-Mean** | **B-Acc** | **G-Mean** | **B-Acc** | **G-Mean** |
| $U = 256$ | $0.7585_{\pm 0.0182}$ | $0.7582_{\pm 0.0128}$ | $0.6821_{\pm 0.0021}$ | $0.6811_{\pm 0.0141}$ | $0.7245_{\pm 0.0361}$ | $0.6894_{\pm 0.0257}$ | $0.7189_{\pm 0.0134}$ | $0.6889_{\pm 0.0179}$ |
| $U = 512$ | $0.7965_{\pm 0.0098}$ | $0.7839_{\pm 0.0378}$ | $0.6870_{\pm 0.0102}$ | $0.6789_{\pm 0.0126}$ | $0.6983_{\pm 0.0184}$ | $0.6379_{\pm 0.0446}$ | $0.7340_{\pm 0.0146}$ | $0.7244_{\pm 0.0274}$ |
| $U = 1024$ | $0.8098_{\pm 0.0104}$ | $0.8014_{\pm 0.0142}$ | $0.6510_{\pm 0.0221}$ | $0.6262_{\pm 0.0300}$ | $0.7411_{\pm 0.0072}$ | $0.7398_{\pm 0.0059}$ | $0.7460_{\pm 0.0071}$ | $0.7362_{\pm 0.0103}$ |
| $U = 2048$ | $0.7972_{\pm 0.0148}$ | $0.7959_{\pm 0.0135}$ | $0.6501_{\pm 0.0229}$ | $0.6332_{\pm 0.0351}$ | $0.7507_{\pm 0.0027}$ | $0.7205_{\pm 0.0169}$ | $0.7247_{\pm 0.0018}$ | $0.7038_{\pm 0.0056}$ |

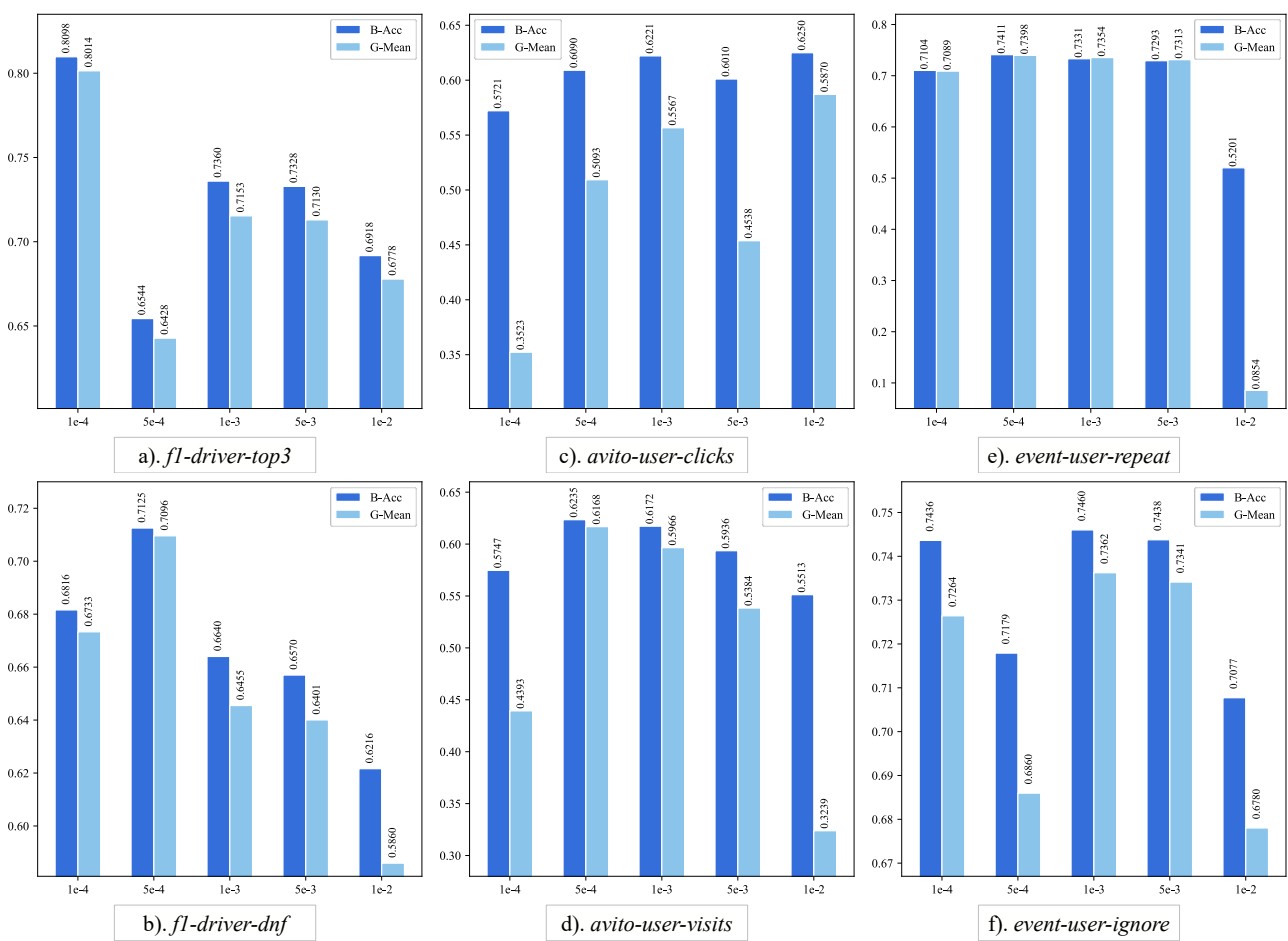

*Figure 8.* Performance of Rel-MOSS versus learning rate $\eta$ on 6 datasets, in terms of balanced accuracy (B-Acc) and G-Mean.

*Table 8.* Entity classification performance of Rel-MOSS versus batch size $B$.

| Dataset | *f1-driver-top3* | | *f1-driver-dnf* | | *event-user-repeat* | | *event-user-ignore* | |
|---|---|---|---|---|---|---|---|---|
| Metric | B-Acc | G-Mean | B-Acc | G-Mean | B-Acc | G-Mean | B-Acc | G-Mean |
| $B = 128$ | $0.6203_{\pm 0.0192}$ | $0.5513_{\pm 0.0120}$ | $0.7002_{\pm 0.0376}$ | $0.6969_{\pm 0.0538}$ | $0.6843_{\pm 0.0255}$ | $0.6824_{\pm \pm 0.0248}$ | $0.7352_{\pm 0.0249}$ | $0.7314_{\pm 0.0364}$ |
| $B = 256$ | $0.8032_{\pm 0.0156}$ | $0.7984_{\pm 0.0261}$ | $0.6050_{\pm 0.0112}$ | $0.5465_{\pm 0.0424}$ | $0.7261_{\pm 0.0282}$ | $0.7190_{\pm 0.0442}$ | $0.7194_{\pm 0.0072}$ | $0.6886_{\pm 0.0137}$ |
| $B = 512$ | $0.8098_{\pm 0.0104}$ | $0.8014_{\pm 0.0142}$ | $0.6510_{\pm 0.0221}$ | $0.6262_{\pm 0.0300}$ | $0.7411_{\pm 0.0072}$ | $0.7398_{\pm 0.0059}$ | $0.7460_{\pm 0.0071}$ | $0.7362_{\pm 0.0103}$ |
| $B = 1024$ | $0.8185_{\pm 0.0124}$ | $0.8138_{\pm 0.0178}$ | $0.6879_{\pm 0.0132}$ | $0.6820_{\pm 0.0189}$ | $0.6576_{\pm 0.0084}$ | $0.6125_{\pm 0.0106}$ | $0.7085_{\pm 0.0073}$ | $0.6714_{\pm 0.0137}$ |

*Table 9.* Entity classification performance of Rel-MOSS with HGT and RelGNN backbones.

| Dataset | *f1-driver-top3* | | *f1-driver-dnf* | | *event-user-repeat* | | *event-user-ignore* | |
|---|---|---|---|---|---|---|---|---|
| Metric | B-Acc | G-Mean | B-Acc | G-Mean | B-Acc | G-Mean | B-Acc | G-Mean |
| RDL-HGT | $0.5124_{\pm 0.0175}$ | $0.1179_{\pm 0.1670}$ | $0.5497_{\pm 0.0324}$ | $0.3647_{\pm 0.1116}$ | $0.6021_{\pm 0.0737}$ | $0.4296_{\pm 0.3045}$ | $0.6820_{\pm 0.0059}$ | $0.6235_{\pm 0.0147}$ |
| Rel-MOSS (HGT) | $0.7675_{\pm 0.0870}$ | $0.7502_{\pm 0.0832}$ | $0.6901_{\pm 0.0458}$ | $0.6857_{\pm 0.0624}$ | $0.6279_{\pm 0.0640}$ | $0.5456_{\pm 0.1788}$ | $0.7125_{\pm 0.0078}$ | $0.6813_{\pm 0.0119}$ |
| RelGNN | $0.4976_{\pm 0.0034}$ | $0.0927_{\pm 0.1311}$ | $0.5272_{\pm 0.0066}$ | $0.2973_{\pm 0.0332}$ | $0.6279_{\pm 0.0671}$ | $0.5822_{\pm 0.1229}$ | $0.6303_{\pm 0.0418}$ | $0.5233_{\pm 0.0869}$ |
| Rel-MOSS (RelGNN) | $0.7439_{\pm 0.0514}$ | $0.7378_{\pm 0.0702}$ | $0.7009_{\pm 0.0486}$ | $0.7008_{\pm 0.0342}$ | $0.6912_{\pm 0.0266}$ | $0.6903_{\pm 0.0262}$ | $0.7341_{\pm 0.0208}$ | $0.7158_{\pm 0.0320}$ |

# D. RDL Backbone

Apart from the standard RDL pipeline based on a heterogeneous GraphSAGE network, we further investigate the effectiveness of Rel-MOSS on two backbones, i.e., the heterogeneous graph transformer (HGT) (Hu et al., 2020) and RelGNN (Chen

et al., 2025). The evaluated results are presented in Table 9. The results demonstrate that Rel-MOSS can deliver consistent improvements in classification performance. The performance gap is most pronounced on the highly imbalanced f1-driver datasets. For instance, on the `f1-driver-top3` task, the vanilla RDL-HGT and RelGNN models yield extremely low *G-Mean* scores (0.1179 and 0.0927, respectively), indicating a severe bias toward majority classes. In contrast, integrating Rel-MOSS boosts these scores to 0.7502 and 0.7378, effectively resolving the imbalance issue. Furthermore, Rel-MOSS demonstrates strong backbone agnosticism. Whether applied to HGT or RelGNN, it consistently improves the discrimination capability of the model, particularly in the `event-user-repeat` and `event-user-ignore` datasets, where it improves *balanced accuracy* by margins ranging from 0.02 to 0.10. This confirms that Rel-MOSS serves as a robust plug-and-play module for enhancing class-imbalanced learning in relational deep learning tasks.

## E. Limitation and Future Direction

In this work, we primarily investigate the imbalance problem of the entity classification task, i.e., class imbalance. However, regarding the entity regression task, the imbalance problem in terms of numerical distribution may negatively impact model optimization. Furthermore, several advanced generative techniques are well-established, including the diffusion model (Rombach et al., 2022) and flow-based model (Lipman et al., 2022). Hence, it is promising to employ an ingenious generative model for minority sample generation. Last but not least, according to common sense, foundation models pre-trained on large-scale datasets might be resistant to the class imbalance problem, although this has not been validated up to now.

## F. Large Language Models Usage

In this work, large language models (LLMs) are employed only in a supporting role. In particular, they were used to enhance the precision of the writing by detecting and correcting grammatical errors and refining word choice, and to propose suitable color palettes for figure design. All research concepts, methodological innovations, experimental work, and the primary text of the manuscript were independently conceived, executed, and authored by the writers.

