# OpenReview forum: "Rel-MOSS: Towards Imbalanced Relational Deep Learning on Relational Databases"
_ICML.cc/2026/Conference — ICML 2026 regular_

### Official Review · Reviewer_xpRs · 2026-03-05

**Soundness:** 4
**Presentation:** 3
**Significance:** 3
**Originality:** 3
**Overall Recommendation:** 4
**Confidence:** 3

**Summary:**

This paper systematically investigates, for the first time, the class imbalance problem in entity classification tasks on relational databases (RDBs). To address the issue that existing relational deep learning (RDL) methods tend to submerge minority entity representations with majority-class information, the authors propose Rel-MOSS, a relation-centric minority synthetic over-sampling GNN framework. The method comprises two core modules: (1) a relation-wise gating controller (Rel-Gate) that adaptively modulates message passing by estimating the minority-leaning likelihood for each relation type; and (2) a relation-guided minority synthesizer (Rel-Syn) that integrates structural statistics of entity neighborhoods as relational signatures into the over-sampling process to preserve relational consistency. Extensive experiments on 12 datasets from the RelBench benchmark demonstrate that Rel-MOSS significantly outperforms existing imbalanced learning methods in terms of Balanced Accuracy (B-Acc) and G-Mean, validating its effectiveness.

**Compliance With Llm Reviewing Policy:**

Affirmed.

**Final Justification:**

I thank the authors for their thoughtful response to the reviews. After carefully re-examining the original paper, the reviewers’ comments, and the authors’ rebuttal, I maintain my original score.

**Key Questions For Authors:**

1. Regarding the proof of Proposition 4.1, there are two issues: (1) How can we compute the expectation of the representation? (2) Concerning formula (8), the statement "Since $\pi_{e,r} \ll 1$ for minority entities and $\|W_{r}^{(l)}\|$ is bounded'' does not guarantee a "contraction mapping''?
2. In formula (9), it appears that the attention weights $(QK^\top)$ are directly multiplied by the value $(V)$, without first applying Softmax normalization. Why is this the case?
3. In the ablation experiments, on some datasets, removing Rel-Gate or Rel-Syn actually yields better performance than the complete Rel-MOSS model. What is the reason for this? Please provide an analysis and explanation.

**Limitations:**

yes

**Strengths And Weaknesses:**

Strengths:
1. The paper presents the first systematic study of the class imbalance problem in entity classification within relational databases (RDB), a prevalent and significant issue in real-world applications.
2. The concept of "relation consistency'' emphasized in the Rel-Syn module profoundly highlights the potential structural bias introduced when performing oversampling in structured RDBs. The proposed mitigation via "relation signatures'' is both ingenious and effective.
3. The paper conducts exceptionally thorough experiments on the RelBench benchmark encompassing 12 datasets, and the breadth of the experimental evaluation is convincing.

Weaknesses:
1. The derivation of Proposition 4.1 employs strong simplifying assumptions (e.g., ignoring non-linearity, using norm upper bounds) to arrive at a concise conclusion. This makes it resemble more of an "intuitive explanation'' than a rigorous mathematical proof.
2. Comparisons with state-of-the-art graph imbalance learning methods (such as GraphSHA, GNN-CL) are only cited in the Related Work section and are not directly compared in the experiments.
3. The writing requires improvement, as there are instances of awkward long sentences that significantly detract from the reading experience and academic rigor.

---

> ### Author Rebuttal · Authors · 2026-03-30
>
> Dear reviewer, thank you for your detailed review. Our point-by-point responses are listed as follows.
>
> > **W1 & Q1**. Proposition 4.1.
>
> - We sincerely thank you for your rigorous mathematical scrutiny. The derivation of Proposition 4.1 is intended as an intuitive theoretical analysis. In GNN literature, employing simplifications, such as Lipschitz constants [1], ignoring non-linearities to focus on topological effects [2], alongside utilizing matrix norm upper bounds [1], is a widely accepted analytical practice to provide theoretical motivation. *(i)* The expectation in our derivation is a theoretical analytical construct used to evaluate the expected trajectory of representations over the data distribution of graph neighborhoods. It is not an empirical value that we explicitly compute during the model forward pass. *(ii)* We appreciate you pointing out this mathematical looseness. We will remove the claim of a strict "contraction mapping." Instead, we will refine the conclusion to state that because $\pi_{e,r}<<1$, the minority discriminative signal undergoes a layer-wise attenuation relative to the dominant majority signal. Even without a strict contraction, this severe attenuation naturally leads to the minority information collapse, which thoroughly motivates the design of our Rel-Gate to actively amplify these decaying signals.
>
> > **W2**. Comparisons with state-of-the-art graph imbalance learning methods.
>
> - Following your constructive comment, we append two SOTA graph imbalance learning methods, i.e., GraphSHA [3] and ReVar [4], as baselines. The evaluated results are partially presented as follows (The complete results is presented in https://anonymous.4open.science/r/Rebuttal-5E1D/ICML_26_Rebuttal.pdf). The evaluated results further demonstrate that previous graph imbalance learning methods designed for homogeneous graph is not directly applicable for heterogeneous relational entity graph.
>
> | **f1-driver-top3** | **B-Acc** | **G-Mean** |
> | :--- | :--- | :--- |
> | GraphSHA | 0.5334 | 0.2836 |
> | ReVar | 0.5579 | 0.3931 |
> | Rel-MOSS | 0.8098 | 0.8014 |
>
> | **f1-driver-dnf** | **B-Acc** | **G-Mean** |
> | :--- | :--- | :--- |
> | GraphSHA | 0.6442 | 0.6179 |
> | ReVar | 0.5975 | 0.5101 |
> | Rel-MOSS | 0.6510 | 0.6262 |
>
> | **event-user-repeat** | **B-Acc** | **G-Mean** |
> | :--- | :--- | :--- |
> | GraphSHA | 0.6904 | 0.6841 |
> | ReVar | 0.7002 | 0.6964 |
> | Rel-MOSS | 0.7411 | 0.7398 |
>
> | **event-user-ignore** | **B-Acc** | **G-Mean** |
> | :--- | :--- | :--- |
> | GraphSHA | 0.6858 | 0.6281 |
> | ReVar | 0.6893 | 0.6366 |
> | Rel-MOSS | 0.7460 | 0.7362 |
>
> > **W3**. Paper writing.
>
> - Thank you for your careful reading and we will keep polishing up our writing.
>
> > **Q2**. Attention weights in formula (9)
>
> - We apologize for the typo here. As you point out, the weight $(QK^T)$ ought to be first normalized by Softmax. We promise to correct this mistake in the final version.
>
> > **Q3**. Analysis on ablation experiments.
>
> - On amazon-user-churn and hm-user-churn datasets, the variant w/o Rel-Syn outperforms Rel-MOSS. As we discussed in the main text, these two datasets are relatively balanced and posses large sample sizes. Hence, the synthetic samples generated by Rel-Syn may potentially introduce negative effect. On f1-driver-dnf dataset, the variant w/o Rel-Gate outperforms Rel-MOSS. This accidental phenomenon implies that for this dataset, preserving all relations are beneficial during the minority over-sampling rather than filtering out some of them.
>
> [1] Graph Neural Networks Exponentially Lose Expressive Power for Node Classification. ICLR 2020.
>
> [2] Deeper Insights Into Graph Convolutional Networks for Semi-Supervised Learning. AAAI 2018.
>
> [3] GraphSHA: Synthesizing Harder Samples for Class-Imbalanced Node Classification. KDD 2023.
>
> [4] Rethinking Semi-Supervised Imbalanced Node Classification from Bias-Variance Decomposition. NeurIPS 2023.

---

> > ### Author Rebuttal · Reviewer_xpRs · 2026-04-03
> >
> > I will keep my score

---

> > > ### Author Response · Authors · 2026-04-03
> > >
> > > Dear reviewer xpRs, thank you for your insightful comments and careful review.

---

### Official Review · Reviewer_E2Aw · 2026-03-13

**Soundness:** 3
**Presentation:** 3
**Significance:** 3
**Originality:** 4
**Overall Recommendation:** 4
**Confidence:** 4

**Summary:**

This paper introduces the Rel-MOSS framework, aiming to mitigate the class imbalance in Relational Database (RDB) entity classification. It addresss challenges such as minority information submergence and unfaithful sample synthesis through adaptive gating and structure-guided synthesis mechanisms.The main contributions of the paper are：

1.Formalized the class imbalance problem in RDBs for the first time and theoretically proved that minority-discriminative signals decay exponentially during standard GNN message passing (termed "minority information collapse")

2.Developed a relation-wise gating mechanism that adaptively modulates message weights based on the minority-leaning likelihood of neighborhoods, enhancing minority signals while restraining majority interference.

3.Proposed an over-sampling method integrating "relational signatures" (e.g. entity type histograms and fan-in/fan-out distributions), ensuring structural faithfulness and logical consistency of synthetic entities within heterogeneous networks.

4.Validated effectiveness across 12 RelBench datasets, achieving average improvements of 2.46% in Balanced Accuracy while 4.00% in G-Mean.

**Compliance With Llm Reviewing Policy:**

Affirmed.

**Final Justification:**

The rebuttal fully solved my concerns.

**Key Questions For Authors:**

1.Appendix E acknowledges a quadratic complexity O(B^2d) for Rel-Gate relative to batch size , yet the conclusion claims no sacrifice in scalability. Will this quadratic overhead become a severe bottleneck for industrial-scale graphs that require massive batch sizes?

2.Experiments show a 25% drop in G-Mean when the dimension increases from 128 to 512, indicating high sensitivity to minority noise. Besides limiting latent dimensions, are there more robust regularization techniques to mitigate this overfitting phenomenon?

3.The core of Rel-Syn relies on local structural statistics. How reliable are these "relational signatures" in real-world RDB environments characterized by poor data quality, such as missing foreign keys or erroneous associations?

4.The current likelihood estimation mechanism is designed for binary classification. How would this gating mechanism distinguish and balance different rare signals in multi-class imbalance scenarios?

**Limitations:**

Yes

**Strengths And Weaknesses:**

**Strengths**

S1:This work is the first to systematically investigate class imbalance in RDBs, filling a critical void in current RDL literature.

S2:Beyond empirical observations, the authors provide a mathematical proof (Proposition 4.1) for "minority information collapse," ensuring the method is theoretically grounded rather than merely heuristic.

S3:Extensive experiments across 12 real-world datasets show significant gains in Balanced Accuracy and G-Mean, while maintaining stability even on non-imbalanced datasets.

**Weaknesses**

W1:The time complexity of the Rel-Gate component is quadratic relative to the batch size (O(B^2d)), which may limit scalability in scenarios requiring massive batches.

W2:Performance is highly sensitive to the latent dimension d and memory bank scale U. For instance, exceeding a dimension of 128 leads to a sharp performance drop due to overfitting.

W3:The research currently focuses only on binary classification and has not yet extended to multi-class scenarios or the imbalance issues prevalent in RDB regression tasks.

---

> ### Author Rebuttal · Authors · 2026-03-30
>
> Dear reviewer, thank you for your detailed review. Our point-by-point responses are listed as follows.
>
> > **W1 & Q1**. Time complexity of the Rel-Gate.
>
> - Regarding the time complexity of Rel-Gate, the quadratic factor $O(B^2)$ comes from the cross attention in Eq.9. Inspired by related research, we want to impart the target entity dependency to the gating process and thus adopt the cross attention between $X_e$ and $H_{e,r}$. To reduce the quadratic time complexity, a reasonable alternative for industrial-scale graphs is the concatenation operation, i.e., $[X_e||H_{e,r}]$. After replacing the cross attention with the concatenation, the performance comparison is presented in table below. The results demonstrate that Rel-MOSS Concat remains comparative to Rel-MOSS. Hence, Rel-Gate itself is not the bottleneck for industrial-scale graphs.
>
> | **f1-driver-top3** | **B-Acc** | **G-Mean** | **PR-AUC** |
> | :--- | :--- | :--- | :--- |
> | RDL | 0.5 | 0.0 | 0.2796 |
> | Rel-MOSS Concat | 0.8068 | 0.8006 | 0.3223 |
> | Rel-MOSS Attention | 0.8098 | 0.8014 | 0.3571 |
>
> | **f1-driver-dnf** | **B-Acc** | **G-Mean** | **PR-AUC** |
> | :--- | :--- | :--- | :--- |
> | RDL | 0.6274 | 0.5835 | 0.2030 |
> | Rel-MOSS Concat | 0.6509 | 0.6319 | 0.2279 |
> | Rel-MOSS Attention | 0.6510 | 0.6262 | 0.2372 |
>
> | **event-user-repeat** | **B-Acc** | **G-Mean** | **PR-AUC** |
> | :--- | :--- | :--- | :--- |
> | RDL | 0.6627 | 0.6377 | 0.7012 |
> | Rel-MOSS Concat | 0.7293 | 0.7292 | 0.7294 |
> | Rel-MOSS Attention | 0.7411 | 0.7398 | 0.7419 |
>
> | **event-user-ignore** | **B-Acc** | **G-Mean** | **PR-AUC** |
> | :--- | :--- | :--- | :--- |
> | RDL | 0.6767 | 0.6136 | 0.4975 |
> | Rel-MOSS Concat | 0.7383 | 0.7298 | 0.5291 |
> | Rel-MOSS Attention | 0.7460 | 0.7362 | 0.5308 |
>
> > **W2 & Q2**. Sensitivity to the latent dimension.
>
> - The sensitivity to latent dimension can be attributed to the dataset sample size. For f1-driver-top3, where the performance drops most as the dimension increasing, the sample size is around 1300 and thus large latent dimension is prone to overfitting. Here we investigate the impact of classical regularizations including $L_1$-norm and $L_2$-norm, the evaluated results are presented as follows. However, the regularization technique has somewhat exceeded our research topic imbalanced RDL. The evaluated results reveal that both $L1$-norm and $L2$-norm can mitigate the overfitting issue under large latent dimension $d$. Meanwhile, we notice that $L2$-norm may degrade the performance of small $d$ and the improvement of $L1$-norm is more consistent.
>
> | f1-driver-top3 | RelMOSS (B-Acc) | RelMOSS (G-Mean) | L1-Norm (B-Acc) | L1-Norm (G-Mean) | L2-Norm (B-Acc) | L2-Norm (G-Mean) |
> | :--- | :--- | :--- | :--- | :--- | :--- | :--- |
> | d=32 | 0.7345 | 0.7274 | 0.7563 | 0.7536 | 0.7175 | 0.7075 |
> | d=64 | 0.7886 | 0.7861 | 0.7916 | 0.7913 | 0.7261 | 0.7201 |
> | d=128 | 0.8098 | 0.8014 | 0.8023 | 0.8012 | 0.7887 | 0.7786 |
> | d=256 | 0.7122 | 0.7122 | 0.7798 | 0.7681 | 0.6935 | 0.6908 |
> | d=512 | 0.6203 | 0.5513 | 0.7220 | 0.7207 | 0.6798 | 0.6616 |
>
> > **W3 & Q4**. Imbalance issues in multi-class scenarios or RDB regression tasks.
>
> - The multi-class imbalance and imbalance in regression tasks are quite meaningful future directions. We deliver related discussion in our limitation and there exist some practical problems in investigating them. First, current RDB entity classification tasks are (mostly) binary classification and the validity of man-made multi-class classification tasks is hard to ensure. Second, the imbalance in regression tasks is more complicated. The definition of 'minority' and 'majority' is unclear. Then, the difference between minority and outliers is also poorly defined. Regarding Q4, theoretically, we can extend Rel-Gate to multi-class scenario by implementing several gating heads, which is similar to the multi-head attention mechanism.
>
> > **Q3**. Real-world RDB environments with by poor data quality.
>
> - The robustness of imbalanced RDL is also a meaningful research problem. The reliability of relational signature mainly depends on the corrupted extent of the RDB. As a neighborhood statistic, relational signature is inherently robust to subtle corruptions. However, the integrity of RDB is a massive research community which beyonds our topic imbalanced RDL.

---

> > ### Author Rebuttal · Reviewer_E2Aw · 2026-04-05
> >
> > I thank the authors for their comprehensive rebuttal and the additional experimental data. The technical concerns regarding the O(B^2d) complexity and high-dimensional overfitting have been successfully addressed through the Rel-MOSS Concat results and the L_1 regularization analysis. These detailed responses have significantly strengthened my confidence in the paper's methodology and made my original assessment of 4 (Weak Accept) even more solid. While the technical doubts are fully resolved, I believe this score remains the most accurate reflection of the work's overall contribution and significance relative to the high standards of ICML. Therefore, I maintain my current score while acknowledging the authors' successful defense of their work.

---

> > > ### Author Response · Authors · 2026-04-05
> > >
> > > Dear reviewer E2Aw, thank you for your and careful review and positive recommendation.

---

### Official Review · Reviewer_8Sbj · 2026-03-13

**Soundness:** 3
**Presentation:** 3
**Significance:** 3
**Originality:** 2
**Overall Recommendation:** 4
**Confidence:** 4

**Summary:**

This paper studies class imbalance in relational deep learning for entity classification over relational databases. The authors propose Rel-MOSS, which contains two components: Rel-Gate, a relation-wise gating mechanism to reweight relational messages during message passing, and Rel-Syn, a minority oversampling method that generates synthetic samples guided by a relational signature summarizing local structural statistics. The method shows improvements on imbalance-aware metrics on the RelBench datasets.

**Compliance With Llm Reviewing Policy:**

Affirmed.

**Final Justification:**

This paper studies a meaningful and practically relevant problem, namely class imbalance in relational deep learning for entity classification over relational databases. By incorporating additional relational modeling and introducing embedding-level minority oversampling, the method demonstrates improved performance on imbalance-related metrics.

My original concerns mainly focused on the similarity of the proposed components to prior methods and the lack of alignment with standard benchmark reporting. After reading the rebuttal, these concerns are largely addressed. The authors clarified the role of Rel-Gate relative to existing attention mechanisms, provided additional results to support their claims, and added further analysis on the relationship between embeddings and relational signatures. While some replies do not cover all datasets, and the precise isolated effects of individual components remain somewhat unclear, the rebuttal sufficiently improves my confidence in the work.

Overall, I am updating my score from 3 to 4. As a final suggestion, I encourage the authors to report results on more datasets using the traditional ROC-AUC metric to better align with prior benchmarks and facilitate comparison with existing work. Even if the method performs slightly weaker under this metric, such results would still be helpful for a more complete understanding.

**Key Questions For Authors:**

1. What is the main difference between Rel-Gate and existing heterogeneous attention mechanisms such as HGT? Could per-relation attention be approximated by existing attention mechanisms, such as per-node attention?
2. Regarding the rel-amazon dataset, where the schema is quite simple and includes only three tables, how can Rel-Gate play an important role in this case?
3. Could you evaluate and compare other baselines using AUC-related metrics (e.g., ROC-AUC or PR-AUC) for a more direct comparison? Alternatively, could you include a simple threshold adjustment for other baselines? These methods are trained under an imbalanced distribution, but some may already have strong ranking ability. I suspect that some models (e.g., RelGNN) might perform significantly better with a simple threshold adjustment, especially since they appear strong in AUC but weaker in Balanced Accuracy.
4. How correlated are the learned GNN embeddings with the relational statistics used in the signature? Intuitively, nodes may have similar GNN embeddings only when their structural statistics are similar, while nodes with similar statistics could still produce different embeddings. It would be helpful if the authors could provide some analysis or visualization to illustrate the relationship between the learned embeddings and the relational signatures, and clarify whether the signature provides information beyond what the GNN embedding already captures.

**Limitations:**

Yes

**Strengths And Weaknesses:**

Strength

This paper addresses a real and practical problem. Class imbalance is common in RDB entity classification tasks such as churn or fraud detection, and the paper clearly explains why standard RDL pipelines may struggle in these settings. The overall framework is also easy to understand, with Rel-Gate aiming to preserve minority-related signals during message passing and Rel-Syn aiming to generate more reasonable minority samples. The experimental results, especially the comparisons with SMOTE-style baselines and the t-SNE visualization of synthetic samples, provide some evidence that the proposed approach is helpful for imbalanced RDL problems.

Weakness

1. The role of Rel-Gate is unclear. The mechanism appears similar to relation-aware attention used in heterogeneous GNNs such as HGT, and it is not clear what new capability it provides. In addition, the motivation is somewhat unclear: label imbalance occurs on target nodes, while useful signals may appear through multi-hop relational patterns rather than through individual relations. The paper lacks an explanation of why distinguishing relation types is helpful, rather than distinguishing neighbors based on their labels or structural roles. Moreover, in simple schemas with only a few relation types (e.g., the rel-amazon dataset with user–purchase–item relations), it is unclear how Rel-Gate can provide meaningful benefits.
2. The interpretation of Rel-Syn raises questions. The synthetic samples appear to exist only in embedding space rather than being inserted into the graph as nodes with edges. As an embedding-level augmentation, the t-SNE visualization would naturally show that the generated embeddings are close to real minority samples. However, it remains unclear whether such samples can truly behave as graph entities during relational reasoning.
3. The experimental evaluation leaves some uncertainty about where the improvements come from. The paper reports Balanced Accuracy and G-Mean but does not include ROC-AUC or PR-AUC. Without ranking-based metrics or threshold-tuned baselines, it is difficult to determine whether the gains come from improved representation quality or simply from better threshold behavior under imbalance.

---

> ### Author Rebuttal · Authors · 2026-03-30
>
> Dear reviewer, thank you for your detailed review. Our point-by-point responses are listed as follows.
>
> > **W1 & Q1 & Q2**. Role of Rel-Gate. Main difference between Rel-Gate and HGT attention mechanism.
>
> - The main purpose of Rel-Gate is to modulate the neighborhood messages based on the sparsity of gating mechanism, which is orthogonal to the attention in HGT layer. We adopt cross attention between $X_e$ and $H_{e,r}$ merely to ensure the gating factor is target-enetity-dependent. The attention in HGT focuses on capturing node-wise importance during the message passing process, which can be combined with Rel-Gate. Regarding Q1, as shown by Table 7 in appendix, Rel-Gate can still boost the performance with HGT as backbone .
> - In RDB entity classification task, only target entities (e.g., drivers) are labeled, most neighbor entities (e.g., races and constructors) themselves are usually neutral in terms of the minority or majority classes. For example, a constructor itself is insufficient to imply the target driver will finish race or not. Hence, we turn to distinguish diverse neighborhoods induced by different relation types. Moreover, comparing the performance of standard RDL and RDL-HGT (Table 1 of the main text), it indicates that introducing node-wise attention to finely distinguish neighbors fails to yield stable improvements.
> - For datasets whose schemas are simple, e.g., rel-amazon, the element-wise gating mechanism in Rel-Gate still serves as an information modulator regarding the neighborhood message. Rel-Gate can basically filter out the noisy signals and maintain the most predictive information.
>
> > **W2**. Interpretation of Rel-Syn.
>
> - First, the embedding-level minority over-sampling methods, such as GraphSMOTE and GraphSHA, are well-established in class-imbalanced learning on graphs. For graph structured data, it is difficult to ensure the validity of newly generated nodes and edges. Second, due to the integrity constraint of RDB, the difficulty of minority over-sampling methods in entity-level is further enlarged. Finally, the minority over-sampling process only happens in training stage. For a well-trained Rel-MOSS model, the inference stage is identical to the standard RDL models.
>
> > **W3 & Q3**. Ranking-based metrics.
>
> - Following your insightful comments, we report the PR-AUC metric of standard RDL, RelGNN, SMOTE, and RelMOSS on four class imbalanced datasets in the table below. (ROC-AUC metric is not credible in class imbalance scenario.) The result indicates that RelMOSS can improve the ranking ability in class imbalanced RDL problem. In summary, Rel-MOSS can not only boost the holistic ranking capacity, but also achieve better classification performance under the default threshold.
>
> | PR-AUC            | RDL      | RelGNN   | SMOTE    | Rel-MOSS  |
> |:-:|:-:|:-:|:-:|:-:|
> | f1-driver-top3    | 0.2796$\tiny{\pm0.0035}$ | *0.3085*$\tiny{\pm0.0149}$ | 0.2993$\tiny{\pm0.0250}$ | **0.3571**$\tiny{\pm0.0071}$ |
> | f1-driver-dnf | 0.2030$\tiny{\pm0.0044}$ | *0.2081*$\tiny{\pm0.0100}$ | 0.1996$\tiny{\pm0.0057}$ | **0.2372**$\tiny{\pm0.0018}$ |
> | event-user-repeat | 0.7012$\tiny{\pm0.0103}$ | 0.6323$\tiny{\pm0.0106}$ | *0.7155*$\tiny{\pm0.0099}$ | **0.7419**$\tiny{\pm0.0024}$ |
> | event-user-ignore | 0.4975$\tiny{\pm0.0087}$ | 0.4525$\tiny{\pm0.0069}$ | *0.5028*$\tiny{\pm0.0161}$ | **0.5308**$\tiny{\pm0.0027}$ |
>
> > **Q4**. Relationship between the GNN embeddings and the relational signatures
>
> - In order to reveal the relationship between the GNN embeddings and the relational signatures, we visualize the pair-wise similarity in terms of embeddings and relational signatures. The figures are presented in the anonymous URL: https://anonymous.4open.science/r/Rebuttal-5E1D/ICML_26_Rebuttal.pdf. In figure 1 and 2, the $x$-axis represents the pair-wise signature similarity and the $y$-axis represents the pair-wise embedding similarity. The visualizations of 4 RDL datasets support our statement that similar GNN embeddings do not necessarily imply similar relational signatures, while nodes with similar signatures tend to have similar embeddings.

---

> > ### Author Rebuttal · Reviewer_8Sbj · 2026-04-04
> >
> > My main concerns are largely addressed, but a few points remain.
> >
> > Q2. It is still somewhat unclear why Rel-Gate is particularly effective in the rel-amazon setting. This dataset contains only a small number of relation types, so the benefit of relation-level gating is not immediately evident. More broadly, the gains from Rel-Gate appear relatively modest even on datasets with richer relational schemas. That said, I do not consider this a primary concern, as Rel-Gate may provide benefits beyond relation diversity, and its effectiveness could depend on dataset characteristics and complexity.
> >
> > W2. I agree that embedding-level oversampling is more practical and easier to implement than generating valid nodes and edges in RDB settings. However, under this design, some of the supporting evidence becomes less convincing. In particular, the t-SNE visualization mainly demonstrates that the generated embeddings are close to real minority samples in latent space (which is expected, since they are derived from embeddings). This does not necessarily justify the stronger claim that these synthetic samples behave as valid relational entities.
> >
> > W3. I am still not fully convinced by the claim that ROC-AUC is not credible under class imbalance. First, ROC-AUC is part of the benchmark evaluation and is reported for all methods, so discarding it entirely seems somewhat arbitrary. If the intention is to highlight its limitations, this should be framed more carefully rather than treating it as invalid. Second, it would be helpful to see whether prior work reports cases where ROC-AUC and PR-AUC exhibit significantly different trends in similar relational or graph imbalance settings, along with a more formal discussion of the conditions under which ROC-AUC becomes unreliable. As stated, this claim appears too strong, especially since prior work such as GraphSMOTE still reports ROC-AUC as a standard metric.

---

> > > ### Author Response · Authors · 2026-04-05
> > >
> > > Dear reviewer 8Sbj, we sincerely appreciate for your elaborate review.
> > >
> > > > Q2. It is still somewhat unclear why Rel-Gate is particularly effective in the rel-amazon setting.
> > >
> > > We agree with your acute insight that the effectiveness of Rel-Gate could depend on dataset characteristics. In detail, it can be analyzed from two aspects. First, by comparing Rel-MOSS with w/o Rel-Gate variant, we notice that Rel-Gate can bring consistent improvements, since the imbalanced class distribution is mitigated by Rel-Syn; thus Rel-Gate can be stably optimized and improve the distinguishability of model. Second, by comparing Rel-MOSS with the w/o Rel-Syn variant, we notice that the model with Rel-Gate is particularly effective on amazon-user-churn and hm-user-churn, where the sample size are large and the imbalance strength are relatively slight (with imbalance ratio as 1.665 and 4.522 respectively). Hence, the influence of class imbalance is not severe in these two datasets.
> > >
> > > > W2. The t-SNE visualization does not necessarily justify the stronger claim that these synthetic samples behave as valid relational entities.
> > >
> > > In the t-SNE visualization, both SMOTE and GraphSMOTE adopt embedding-level oversampling methods (GraphSMOTE attempts to generate edges in addition). Since they fail to take relational information into account, the synthetic samples severely deviate from the true samples. Hence, the faithful embedding distribution is not a natural outcome induced by embedding-level oversampling. The goal of the t-SNE visualization is to demonstrate that our synthetic samples are faithful to the real manifold of minority samples. Therefore, the risk of synthetic samples disturbing optimization process is reduced.
> > >
> > > > W3. I am still not fully convinced by the claim that ROC-AUC is not credible under class imbalance.
> > >
> > > We apologize for our non-rigorous over-statement that ROC-AUC is not credible under class imbalance. We promise to provide a careful discussion on the ROC-AUC metric and the adopted metrics in the final version. First, our original intention is to faithfully reflect the performance of different methods under class imbalance scenarios. Hence, we adopt two metric, i.e., b-Acc and G-Mean, which are customized to the class-imbalance problem while seldom used in current literature [1]. Second, during our investigation, [2] reveals that *a curve dominates in ROC space if and only if it dominates in PR space*. Therefore, we select the PR-AUC score, which is more informative in class-imbalanced problems, as the our supplemented metric. Finally, we present the ROC-AUC score as follows, in order to provide a comprehensive evaluation. We notice that all the evaluated methods exhibit competitive ROC-AUC scores which are significantly larger than 0.5 and the difference between models are relatively subtle.
> > >
> > > |      ROC-AUC      |    RDL   |  RelGNN  |   SMOTE  |  Rel-MOSS |
> > > | :---------------: | :------: | :------: | :------: | :------: |
> > > |   f1-driver-top3  | 0.7572 | *0.7792* | 0.7369 | **0.7999** |
> > > |   f1-driver-dnf   | *0.7091* | 0.6911 | 0.7017 | **0.7153** |
> > > | event-user-repeat | 0.7674 | 0.7212 | *0.7808* | **0.7959** |
> > > | event-user-ignore | 0.8092 | 0.8301 | *0.8318* | **0.8524** |
> > >
> > >
> > > [1] Class-Imbalanced Learning on Graphs: A Survey. ACM Comput. Surv. 2025.
> > >
> > > [2] The Relationship Between Precision-Recall and ROC Curves. ICML 2006.

---

### Official Review · Reviewer_f6K5 · 2026-04-07

**Soundness:** 3
**Presentation:** 3
**Significance:** 2
**Originality:** 2
**Overall Recommendation:** 4
**Confidence:** 4

**Summary:**

This paper introduces a learning framework for relational deep learning that mitigates label imbalance issues. Its main novelty is two special designs for heterogeneous graphs, a gating module that predicts the label imbalance in the neighborhood, and signature-based synthesizer for oversampling minority-class nodes. Experiments show significant improvement over baselines.

**Compliance With Llm Reviewing Policy:**

Affirmed.

**Final Justification:**

na

**Key Questions For Authors:**

Please see weaknesses.

**Limitations:**

Yes.

**Strengths And Weaknesses:**

Strengths:
1. Overall this is a well-written paper presenting a well-finished work. The studied problem is important, the method is carefully designed, and the experiments show significant improvement.
2. The specific adaptation of signature-based synthesizer for heterogeneous graphs is novel and reasonable.
3. Experiments are very substantial, covering many datasets and paired with detailed analysis.

Weaknesses:
1.  It seems that this paper is actually proposing a more general method for handling class imbalance in heterogeneous graphs, not just RDBs? If so, I think there are already several works [1-3] that studies this topic (class imbalance in heterogeneous graphs). So the claim in the last paragraph of Sec. 2 is not true? The authors should mention about these methods and highlight the uniqueness of this work compared with them.
2. Have the authors studied how accurate the Rel-Gate is in estimating the likelihood after co-trained it with other objectives?  Although this is a reasonable module to me, it inevitably introduces some error in estimation that can be propagated to downstream. I don't seem to have seen a specific loss designed to guardrail this module's behavior, so I think this paper would benefit from more analysis on this module in experiments (or even consider creating a specific loss for this?).


[1] Fincgan: A gan framework of imbalanced node classification on heterogeneous graph neural network
[2] Heterogeneous graph neural network with relation-aware label propagation for unbalanced node classification
[3] Shine: A scalable heterogeneous inductive graph neural network for large imbalanced datasets

---

### Decision · Program_Chairs · 2026-04-30

**Decision:**

Accept (regular)

**Comment:**

This paper investigates the critical and frequently overlooked issue of class imbalance in Relational Deep Learning (RDL). The authors introduce Rel-MOSS, which utilizes a relation-wise gating mechanism and a structure-preserving synthetic over-sampling strategy to maintain minority class signals within complex relational databases. The work effectively fills a gap between traditional imbalanced learning and the emerging field of RDL.

Strengths:
1. The research addresses a high-impact practical problem, particularly relevant for industrial RDL applications like fraud detection where label distributions are naturally skewed.
2. The proposed framework is technically well-grounded, demonstrating superior performance over existing GNN baselines on the RelBench benchmark while respecting the unique structural constraints of RDBs.
3. The methodological design, moving beyond generic over-sampling to a relation-centric approach, shows strong potential for real-world deployment.

Weaknesses:
1. Some reviews raised valid concerns regarding the overlap between the proposed gating mechanism and standard heterogeneous attention, as well as the method's generalizability across diverse database schemas.
2. The authors should consider conducting a more comprehensive evaluation using ranking-based metrics (e.g., PR-AUC) and a clearer justification for the effectiveness of synthetic samples generated within the latent space.

Overall:
The authors provided a convincing rebuttal that addressed the primary concerns through additional experiments and technical clarifications. Given the positive consensus following the discussion phase, I recommend the paper for acceptance. Please ensure the final version incorporates the new metrics and discussions from the rebuttal.